# Modeling Emergency Department crowding: Restoring the balance between demand for and supply of emergency medicine

**John Pastor Ansah**[1], **Salman Ahmad**[1], **Lin Hui Lee**[2], **Yuzeng Shen**[3], **Marcus Eng Hock Ong**[1,3], **David Bruce Matchar**[1], **Lukas Schoenenberger**[4]*

1 Programme in Health Services and Systems Research, Duke-NUS Medical School, Singapore City, Singapore, 2 Operations & Performance Management, Singapore General Hospital, Bukit Merah, Singapore City, Singapore, 3 Department of Emergency Medicine, Singapore General Hospital, Bukit Merah, Singapore City, Singapore, 4 Department of Health Professions, Institute of Health Economics and Health Policy, Bern University of Applied Sciences, Bern, Switzerland

* lukas.schoenenberger@bfh.ch

**Data Availability Statement:** All relevant data are within the paper and its Supporting Information files.

## Abstract

Emergency Departments (EDs) worldwide are confronted with rising patient volumes causing significant strains on both Emergency Medicine and entire healthcare systems. Consequently, many EDs are in a situation where the number of patients in the ED is temporarily beyond the capacity for which the ED is designed and resourced to manage—a phenomenon called Emergency Department (ED) crowding. ED crowding can impair the quality of care delivered to patients and lead to longer patient waiting times for ED doctor's consult (time to provider) and admission to the hospital ward. In Singapore, total ED attendance at public hospitals has grown significantly, that is, roughly 5.57% per year between 2005 and 2016 and, therefore, emergency physicians have to cope with patient volumes above the safe workload. The purpose of this study is to create a virtual ED that closely maps the processes of a hospital-based ED in Singapore using system dynamics, that is, a computer simulation method, in order to visualize, simulate, and improve patient flows within the ED. Based on the simulation model (virtual ED), we analyze four policies: (i) co-location of primary care services within the ED, (ii) increase in the capacity of doctors, (iii) a more efficient patient transfer to inpatient hospital wards, and (iv) a combination of policies (i) to (iii). Among the tested policies, the co-location of primary care services has the largest impact on patients' average length of stay (ALOS) in the ED. This implies that decanting non-emergency lower acuity patients from the ED to an adjacent primary care clinic significantly relieves the burden on ED operations. Generally, in Singapore, there is a tendency to strengthen primary care and to educate patients to see their general practitioners first in case of non-life threatening, acute illness.

## 1. Introduction

Emergency Departments (EDs) worldwide have to deal with rising patient volumes causing significant pressures on both Emergency Medicine (EM) and entire healthcare systems [1–3].

**Funding:** The authors received no specific funding for this work.

**Competing interests:** The authors have declared that no competing interests exist.

Therefore, many EDs are in a situation where the number of patients occupying the ED is temporarily beyond the capacity for which the ED is designed and resourced to manage—a phenomenon called *Emergency Department (ED) crowding* [4–6]. Particularly, ED crowding can lead to (i) reduced quality of care, (ii) longer patient waiting times for doctor's consult (time to provider), (iii) increased numbers of patients who leave without being seen, and (iv) more ambulance diversion [5].

ED crowding has financial implications causing costs per patient to rise because the average (inpatient) length of stay can be extended [7, 8]. Furthermore, it is assumed that inadequate care due to ED crowding might increase the probability of being readmitted to the ED which further contributes to rising health care costs [9]. Since patients increasingly use EDs as point of entry into the health care system, ED crowding is not only an EM specific nuisance but rather a public health problem [10]. Due to the relevance of ED crowding and the pressures it causes on healthcare systems, a remarkable number of studies on the topic have been published in the Operational Research (OR) literature recently [11–13].

Singapore is no exception to the international trend of rising ED attendance and crowding [3, 14, 15]. In 2016, the total population of Singapore amounts to 5.61 million with an average annual growth rate of 1.3%. In comparison, total ED attendance at public hospitals has grown at a disproportionately higher rate, that is, roughly 5.57% per year between 2005 and 2016 [16]. Fig 1 illustrates the evolution of both total ED attendance at public hospitals and population size in Singapore for the period from 2005 to 2016.

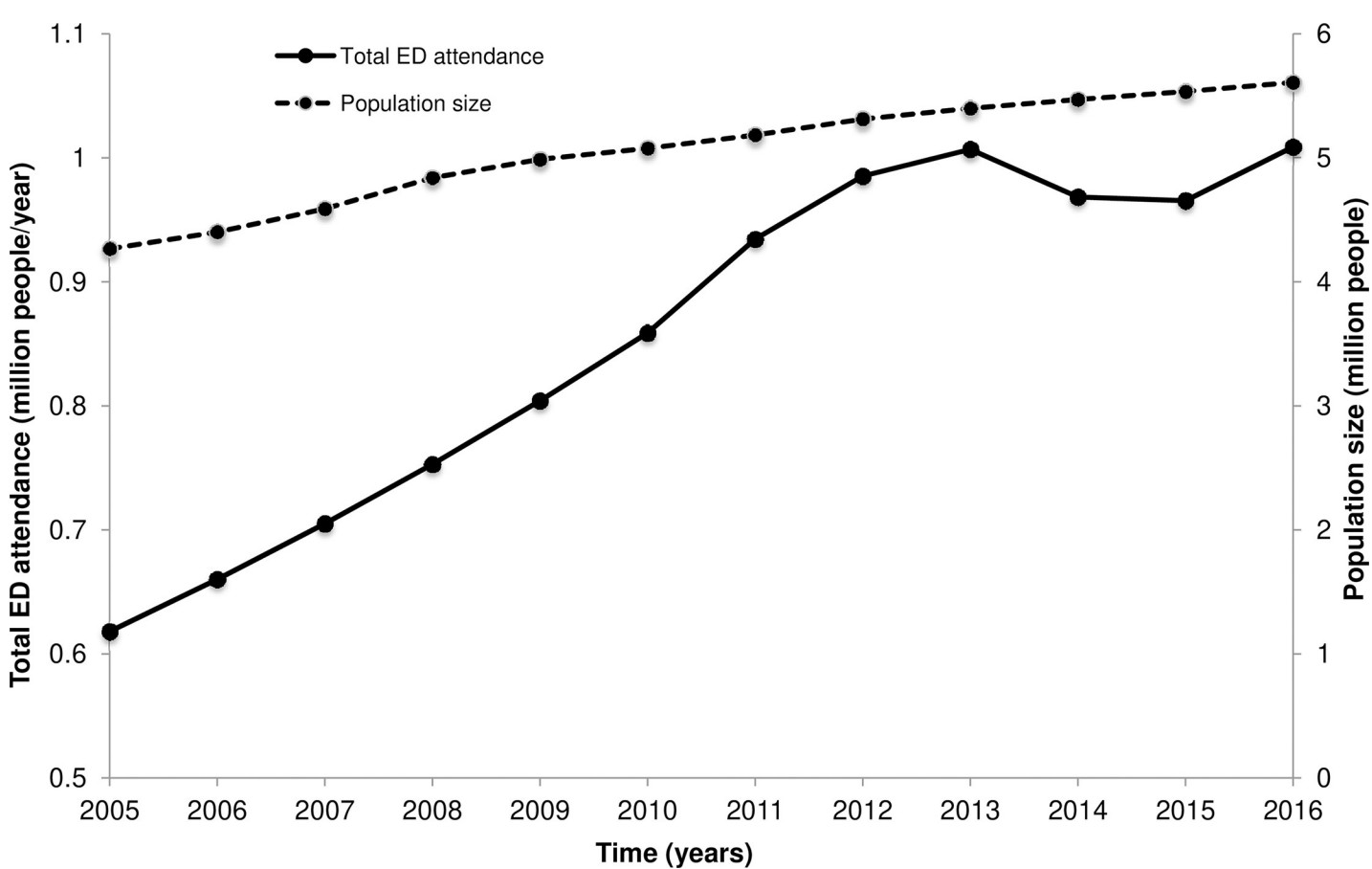

**Fig 1. Evolution of population size and total ED attendance in Singapore from 2005 to 2016.**

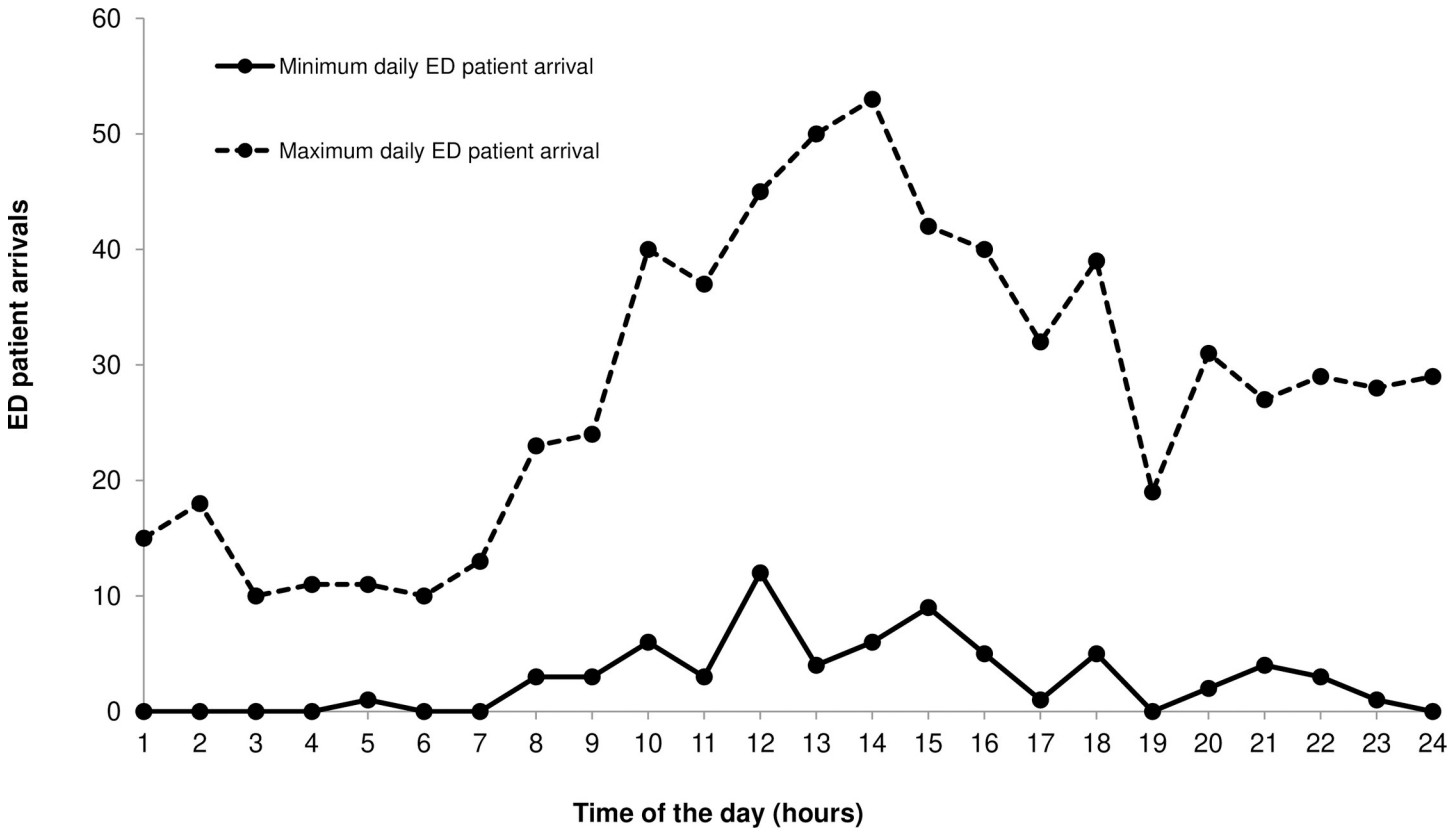

**Fig 2. Patient arrivals over the course of a peak day and a 'quiet' day in a hospital-based ED in Singapore.**

Furthermore, EDs in Singapore and worldwide must cope with highly variable patient arrivals. Typically, patient arrival patterns are cyclic both during the course of a day and over the course of a week [17]. This variable demand and the fact that patient arrivals are unpredictable and stochastic pose an additional burden on ED management teams. Fig 2 emphasizes the variability of patient arrivals by displaying the arrival patterns over the course of a peak day and a 'quiet' day in an ED in Singapore in one month. Although the number of emergency physicians (EPs) has risen, that is, 13.4% annually between 2005 and 2014, their workload has remained very high. In Singapore, on average, each EP sees between 6.4 and 8.5 patients per hour depending on the mode of calculation while previous studies have suggested that the optimal ED throughput lies between 2 and 2.8 patients per EP hour [14, 18]. Therefore, it can be assumed that there is still an undersupply of ED personnel and significant investments into training must be made. Considering rising patient numbers and the workload of ED staff, the current state of emergency medical care in Singapore might not be sustainable and has consequences for the well-being of both patients and ED professionals [3].

The purpose of this paper is to develop a *virtual ED*, i.e., a simulation model that comprehensively reflects all major patient flows and medical resources of a hospital-based ED in Singapore, that is fully transparent (documented) and accessible for researchers and subject experts. Subsequently, the virtual ED is used to analyze the effectiveness of currently debated policies to streamline ED operations in Singapore. Specifically, we investigate the impact of (i) co-location of primary care services within the ED, (ii) increase in the capacity of doctors, (iii) a more efficient patient transfer to inpatient hospital wards, and (iv) a combination of policies (i) to (iii), on patients' average length of stay (ALOS) in the ED. To that end, we use system

dynamics (SD), an advanced simulation modeling approach that is currently underutilized in the modeling of ED operations. SD is a handy approach in this context because its main modeling elements, the so-called 'stocks' and 'flows', make it particularly easy to model aggregate patient flows and stock of patients in a health care setting. Consequently, SD is a useful approach to assess patient flow optimizing policies in an ED.

There are only a handful of studies that analyze ED processes through an SD lens, indicating a gap in the literature (a more thorough discussion of previous modeling works follows in the literature review section). [19] developed a model to simulate the effect of point-of-care testing on ED crowding but the model has not been made publicly available. [20] focused on a specific subgroup of ED patients, that is, patients that were later admitted to general internal medicine in the hospital. Similarly, the model has not been made available. [21] modeled the interplay between an ED and the associated hospital wards focusing on the trade-off between emergency admissions and elective admissions. Unfortunately, the precise sub-model referring to the processes within the ED has not been made transparent and so cannot be evaluated. [22] created a model of hospital patient flows with the aim to define policies that reduce delays within the ED. The model has not been made available. [23] and [24], two related studies, had a broader perspective and modeled emergency care systems instead of detailing a single ED. The respective models have not been made open-access. Finally [12], studied the acute bed blockage problem in the Irish healthcare system but refrained from modeling patient flows within the ED. The model is not available.

To the best of our knowledge, there is no study using SD to create a virtual ED as we understand the term—a comprehensive representation of all major patient flows and corresponding medical resources in an ED—which is thoroughly documented and open-access. Complete model transparency and free access, however, are crucial if models shall be refined, validated, and reused by others. For that reason, in this paper, we put great emphasis on listing and explaining all model equations that are necessary to rebuild the simulation model. Furthermore, because many EDs are structured similarly having critical care (resuscitation care), isolation care, and ambulatory care areas [25], the model we present here can quite easily be translated into any hospital-based ED worldwide [26]. (Currently, we are adapting the model to fit to the largest ED in Switzerland.)

## 2. Literature review

The public importance, the wait-for-treatment ethos and the clear structural layout of EDs have contributed to them being one of the most commonly modeled systems in OR healthcare [25]. A recent and comprehensive literature review on simulation modeling methods applied to EDs identified in total 254 relevant publications, of which 209 used discrete event simulation (DES), 25 agent-based simulation (ABS), 18 SD, and 2 other modeling approaches [25]. The dominance of DES in ED modeling seems justified considering the method's strengths in handling individual patient flows and random variation of variables [27, 28]. We do not deny the suitability of DES in modeling ED processes and crowding. However, instead of focusing on DES alone, we argue for a diversity of simulation methods to be applied to ED operations. In our opinion, only such a multi-perspective approach can lead to new insights. In the following, we limit ourselves to reviewing the seven SD studies briefly touched upon in the introduction. Among the 18 SD works, we selected those of high-quality, written in English, and published in renowned international OR emergency medicine journals.

[19] developed an SD model to examine the effect of decreasing lab turnaround time on emergency medical services diversion, ED patient throughput, and total ED length of stay (LOS). Unfortunately, there is no information on model conceptualization. They concluded

that compelling improvement in ED efficiency with decreasing lab turnaround time can be attained. [20] constructed an SD model to study the impact of evenly distributing inpatient discharges over the course of a week on the bed occupancy rate. The model is limited to only include ED patients that are later admitted to general internal medicine (GIM) in the associated hospital. Model conceptualization entails three main components: (i) a patient category component, (ii) a hospital location component, and (iii) a feedback mechanism component. The interplay between the three components steers the movement of patients from hospital admission to discharge. They found that discharging patients evenly across the week can significantly reduce bed requirements and ED LOS.

[21] built an SD model to analyze the response of ED waiting times to reductions in bed capacity. To that end, they conceptualized the system in terms of two areas: (i) the community, and (ii) the hospital which is further subdivided into the ED, the management of elective patients, and the wards. The simulation model was subsequently used to assess the impact of changes in bed capacity and in ED demand on various key performance measures. The key finding was that reductions in bed numbers do not increase waiting times for emergency admissions because elective admissions fall sharply. So, the elective cancellation rate acts as a so-called 'safety valve' compensating for any change in bed capacity. [22] developed a hospital-wide SD model to improve understanding of the causes of delays and length of stay variations experienced by patients in the ED. They tested the impact of altering nurse levels, delay reductions, and re-routing of patients on total ED length of stay, particularly for admitted patients. Overall, however, the main purpose of this study was to evaluate the applicability of SD to patient flow modeling. It was concluded that the quantitative approach to simulating ED delays and patient flows using SD is reasonable and that the resulting model is appropriately representative of the system under consideration.

[23] and [24] adopted an SD modeling approach to describe the components of an emergency and urgent care system and to investigate ways in which patient flows and system capacity could be improved. The developed model was then used to test the effect of changes in emergency/elective admissions, 'front door' demand, patient discharge schemes, and bed capacity. They found that strengthening community care has the greatest potential to relieve pressure on the emergency and urgent care system.

Finally, [12] created an SD model that visualizes and simulates the dynamic flow of elderly patients in the Irish healthcare system to better understand the system's dynamic complexity, i.e., the nonlinear interactions of system elements over time. The model focuses on general patient pathways of emergency admissions through the entire Irish healthcare system. Special emphasis is placed on post-acute care by including long-term care, care at home, convalescent care, and rehabilitation care in the model. Based on the simulation model, they evaluated various pre-acute, e.g., increasing general practitioners' (GPs) access to community services, and post-acute, e.g., increasing discharge rates from long-term care facilities, policy interventions. They found that a mixed strategy of pre-acute and post-acute policy interventions is potentially very effective in reducing pressures on acute care provision.

Based on this literature review, although not systematic, it can be said that the simulation model presented herein is the first attempt to model all relevant patient flows running through critical care, ambulatory care, and isolation care of a large interdisciplinary hospital-based ED using SD. The novelty of this work does not lie in the particular case study selected here (ED of the largest tertiary hospital in Singapore) but on the detailed and comprehensive representation of all major ED patient flows in an aggregated form. Furthermore, and equally important, the model presented below is described in such a way that interested parties can rebuild, test, and experiment with the model increasing the value of our work.

## 3. Study setting

### 3.1. Methods

SD is a computer-facilitated approach to policy analysis and design with a focus on modeling stocks (accumulations) and flows (rates) of systems. Typically, SD is applied to dynamic problems that are characterized by interdependence and mutual interaction of elements, information feedback, and circular causality [29, 30]. Virtual worlds, i.e., simulation models, created with SD can act as learning laboratories with the purpose of developing and testing strategies before they are implemented in practice. This is highly relevant for organizations nowadays considering the fact that many of them operate within increasingly dynamic environments and, therefore, strategies have to be evaluated and adjusted constantly [31, 32].

We chose SD as our modeling approach for the following reasons. First, it seemed important to us that ED operations are not only analyzed from one methodological viewpoint, that is, discrete event modeling, but tackled by a diversity of simulation methods in order to generate new insights. Second, agreeing with [21], we think that considering aggregated variables (e.g., aggregated flows of patients) which is the focus of SD encourages both a systemic view of the interactions of patient flows and information, and a more strategic perspective of the management of the system. Third, due to its accessible graphical iconography, SD is particularly useful to engage stakeholders both in the model building and in the model analysis phase [33]. In SD, the model structure can be explained and presented in simple mathematical terms which facilitates communication with a non-technical audience. Additionally, SD models take high-level policies as inputs making them accessible for interpretation and fostering dialogue between hospital stakeholders and the modeling team. This was a key aspect to us because we intended to involve EPs, nurses, and ED managers throughout the entire modeling process. SD is still our method of choice when it comes to stakeholder involvement, despite recent efforts in facilitated discrete-event simulation modeling [34].

### 3.2. ED under study

Singapore is a city state with a population of 5.61 million people [16]. The study institution is the largest hospital in the country with 1'600 inpatient beds and provides tertiary care to a significant share of the population. The hospital is part of the Singhealth Regional Health System (RHS) which covers a population of more than 1.1 million people and handles more than 4 million patient visits yearly. The hospital-based ED cares for more than 140'000 patient visits annually, with about 350 visits per day in 2019 [35, 36]. The ED is equipped with 25 specialist EPs who work an average of 180 clinical shift hours per 28 days, along with roughly 40 non-specialists who clock an average of 216 clinical shift hours in the same period [35].

### 3.3. Overall structure of the ED

Patients come to the ED by ambulance or other forms of transportation (walk-in) from the community to seek care. Upon arrival, ED patients go through a brief registration before the triage processes commence. Triage refers to the categorization of ED patients for treatment in situations of scarce resources according to the patients' medical conditions and established sorting plan. In Singapore, the Patient Acuity Category Scale (PACS) which prioritizes patients into four main priorities is used to triage patients at the ED [37]. The priorities are: (1) Priority 1 are patients in a state of cardiovascular or imminent collapse. They are the most serious, time-critical patients who require immediate attention or resuscitation—examples of conditions are heart attack, severe injuries, severe bleeding, shock and severe asthma attack; (2) Priority 2 patients are non-ambulant patients with acute medical conditions who appear to be in

a stable state with no immediate danger of collapse—examples of conditions are major limb fracture/dislocation, moderate injuries, severe abdominal pain and other severe medical illnesses; (3) Priority 3 refers to ambulant patients with acute symptoms who are in a stable condition. These patients could be treated by general practitioners, family physicians with acute care resources—example of conditions are sprains, minor injuries, minor abdominal pain, vomiting, fever, rashes and mild headaches; Finally, (4) Priority 4 are non-emergency patients with old injuries or conditions that have been present for a long time—examples include chronic joint pain, chronic skin rash, long-term nasal discharge, old scars, cataracts, removal of tattoos and sore throats.

A trained nurse evaluates the patient's condition, takes his or her medical history, initiates diagnostic measurements, and determines the priority for treatment, i.e., P1, P2, P3 or P4. Patients with fever, irrespective of treatment priority—P1, P2, P3, or P4—are sent to the *isolation area* to be seen by a physician to reduce the risk of infecting other patients in the ED. Non-ambulant or trolley-based patients in priority 1 and 2 are treated at the *critical care area*, while ambulatory patients, irrespective of their treatment priority are sent for treatment at the *ambulatory care area*.

Each treatment area—critical care, ambulatory care, and isolation care—has a dedicated waiting area and allocated ED nurses and physicians. The average waiting time to consult an ED physician depends on the number of ED patients waiting for consultation and the number of ED physicians available. The higher the ED patient's acuity or priority, the greater the average physician consultation time. For P1 and P2 patients receiving treatment in the critical care area, after initial consultation, almost all patients are admitted to the observation ward for observation. During observation, patients who require laboratory services undergo the investigation there and wait for the results. If there are no beds in the observation ward, patients are observed in the waiting area. For ambulatory patients, after initial consultation, laboratory services are provided. Those who require observation are admitted into the observation ward, while others wait for laboratory investigation results in the waiting area. Lastly for isolation patients, after initial consultation, patients are observed before a decision to discharge is made.

For patients in the observation ward, a decision to send them home or admit them into the hospital is made after a further review by ED physicians. To that end, laboratory results (if available) are reviewed. Discharged ED patients proceed to the pharmacy for medication and payment. For those who require hospital admission, arrangement is made with the appropriate hospital ward for patient transfer. An overview of the principal processes in a hospital-based ED in Singapore is shown in Fig 3.

### 3.4. Model structure

SD models consist of an interconnecting set of differential and algebraic equations developed from a broad range of empirical data. SD models comprise of stocks, interconnected flows and auxiliary variables. A general mathematical representation of stocks and flows are:

$$Stock(t) = \int_{t_o}^{t} [inflow(t) - outflow(t)]dt + Stock(t_o) \tag{1}$$

$$Inflow(t) = f(Stock(t), N) \tag{2}$$

$$Outflow(t) = f(Stock(t), M) \tag{3}$$

where *N* and *M* are the system parameters. The flows are the derivatives or rates of change of the associated stocks. Stocks create disequilibrium dynamics as they decouple flows. As a consequence, typically, inflows and outflows differ and are governed by different decision rules.

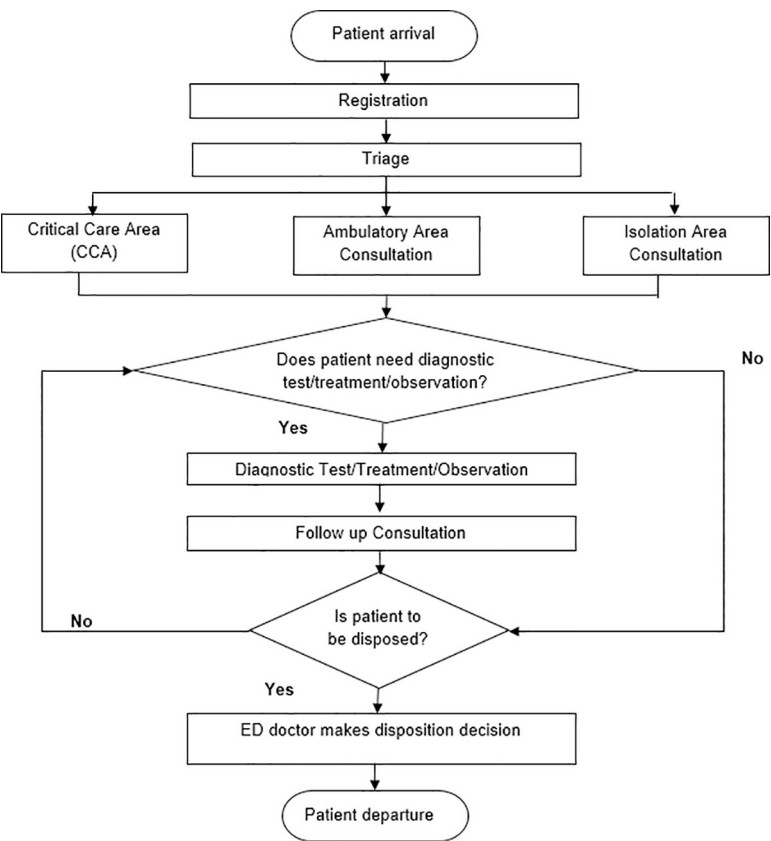

**Fig 3. Overall structure of patient processing in a hospital-based ED in Singapore.**

The overall model structure of an ED in Singapore is presented in Fig 4. For a list with all variables and their respective abbreviations see S2 Table.

**3.4.1. Registration and triage.** For ED patients, the journey begins when they arrive at the ED to seek care (see Fig 5). New patient arrivals $a(t)$ at any time ($t$) proceed for registration and quickly transition from registration to triage. The equation for patients waiting for registration $P(t)$ at time ($t$) is:

$$P(t) = \int_{t_0}^{t} [a(t) - g(t)]dt + P(t_0) \tag{4}$$

where $P(t_0)$ is patients waiting for registration at time ($t_0$) and

$$a(t) = exogenous\ data \tag{5}$$

$$g(t) = [a(t)(t - RT)] \tag{6}$$

New patient arrivals $a(t)$ is an exogenous input and is fed into the simulation model as historical time-series data; $g(t)$ is patients moving from registration to triage and is represented herein as a pipeline delay function of new patient arrivals $a(t)$ and average registration time $RT$.

After registration at the ED, patients wait to be triaged. Patients are normally triaged into four treatment priorities ($j$)—P1, P2, P3, and P4; and three care areas—critical care, ambulatory care, and isolation care. $B(t)$, that is, patients waiting for triage, increases as patients move from registration to triage $g(t)$ and decreases as patient are triaged to critical care $cca_j(t)$,

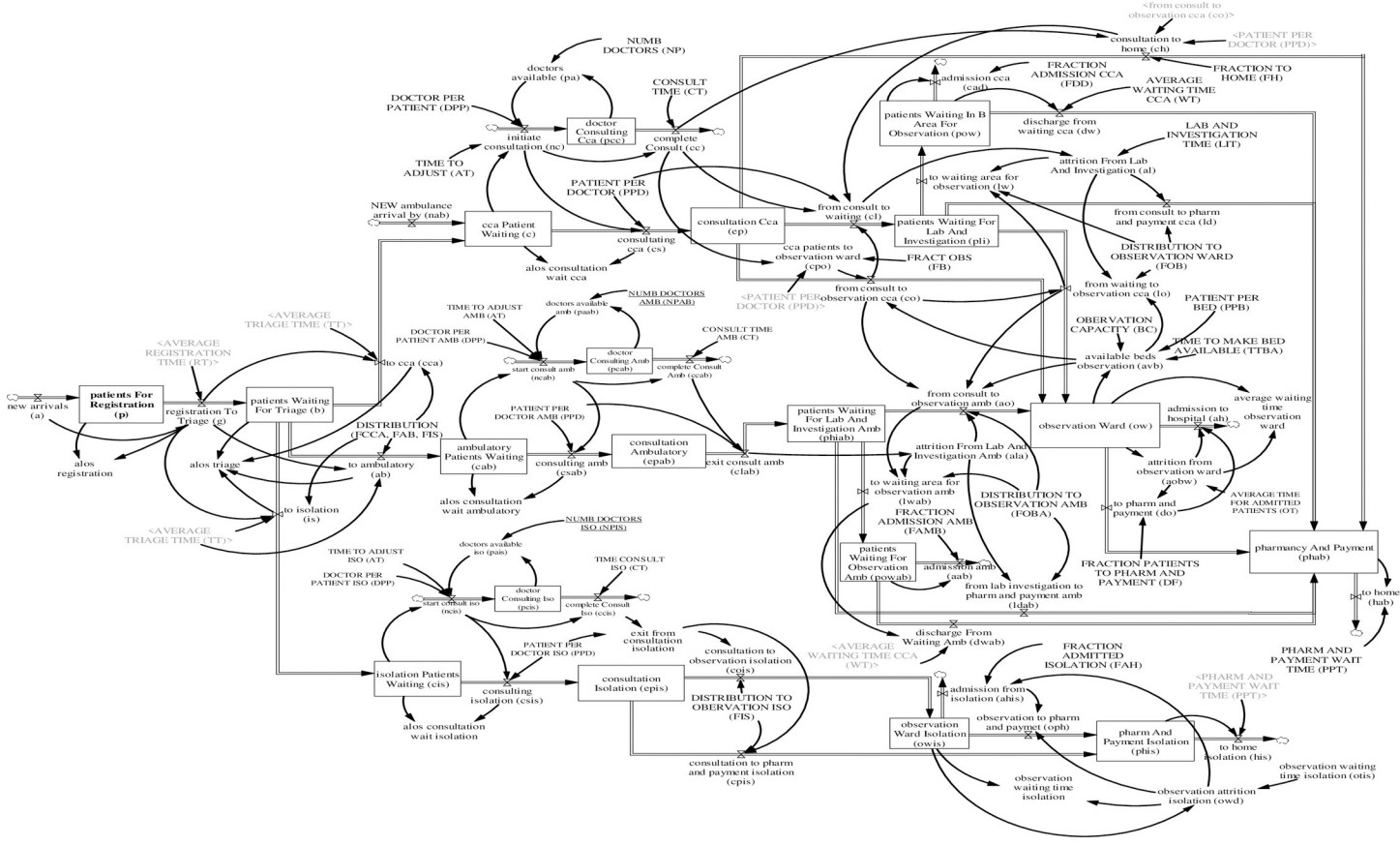

**Fig 4. Overall model structure of an ED in Singapore.**

ambulatory care $ab_j(t)$, and isolation care $is_j(t)$. The equation for patients waiting for triage $B$ $(t)$ is:

$$B(t) = \int_{t_0}^{t} [g(t) - cca_j(t) - ab_j(t) - is_j(t)]dt + B(t_0) \qquad (7)$$

where $B(t_0)$ is patients waiting for triage at time $(t_0)$ and

$$cca_j(t) = g(t - TT) * fcca_j(t) \qquad (8)$$

$$ab_j(t) = g(t - TT) * fab_j(t) \qquad (9)$$

$$is_j(t) = g(t - TT) * fis_j(t) \qquad (10)$$

Patients triaged to critical care $cca_j(t)$, ambulatory care $ab_j(t)$, and isolation care $is_j(t)$ are modeled herein as pipeline delay functions of patients moving from registration to triage $g(t)$ and average triage time $TT$, adjusted by the fraction of patients sent to each care area; $fcca_j(t)$ is the fraction of patients triaged to critical care, $fab_j(t)$ is the fraction of patients triaged to ambulatory care, and $fis_j(t)$ is the fraction of patients triaged to isolation care. All three fractions sum up to one.

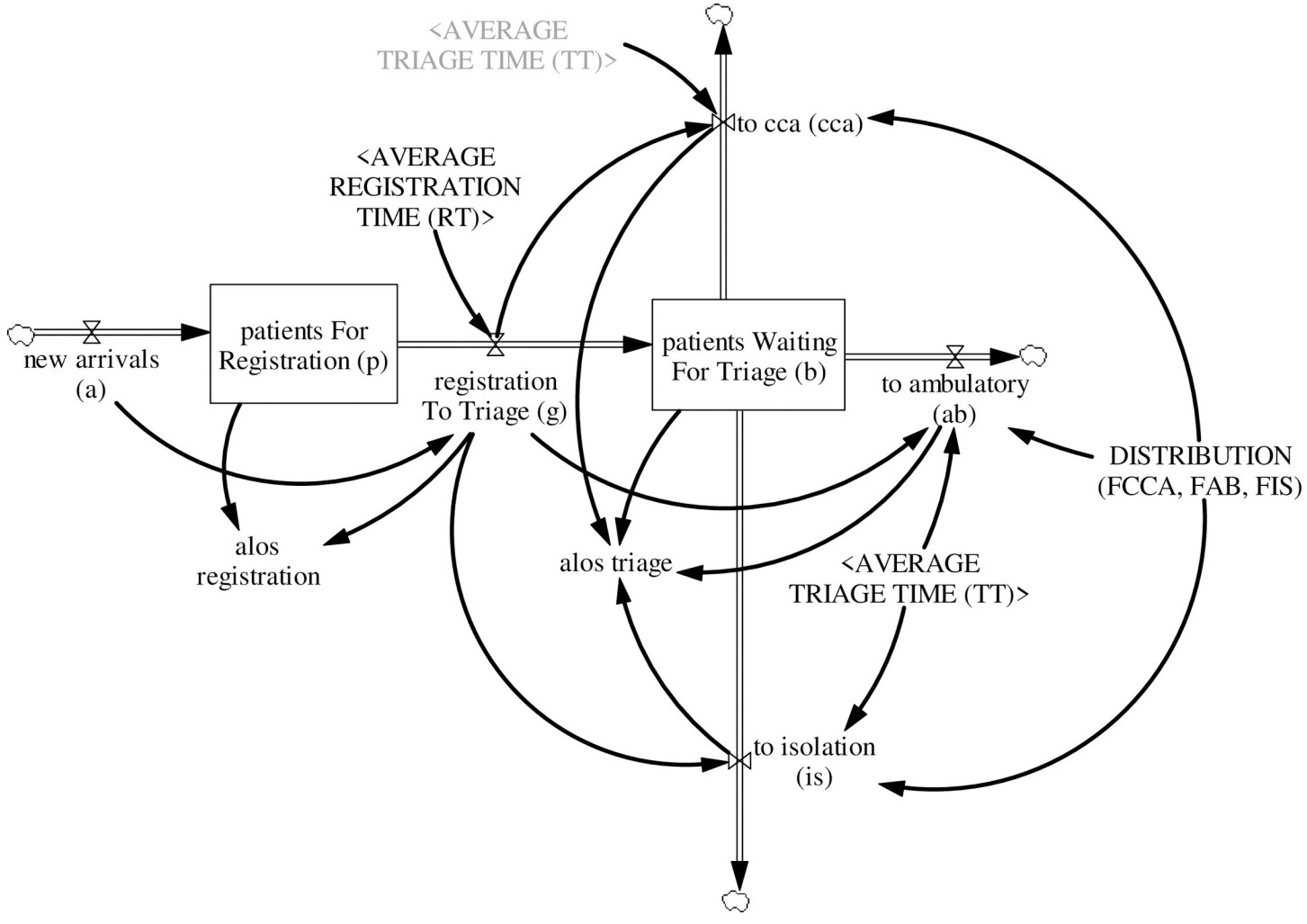

**Fig 5. Registration and triage sub-model.**

**3.4.2. Critical care pathways.**  Patients triaged to the critical care area wait in queue for consultation. In total, there are four main patient pathways within the critical care area (see Fig 6):

(i)  Waiting for consultation → consultation → discharge

(ii)  Waiting for consultation → consultation → laboratory investigation → discharge

(iii)  Waiting for consultation → consultation → observation → discharge

(iv)  Waiting for consultation → consultation → laboratory investigation → observation → discharge

The number of patients waiting for consultation $C_j(t)$ increases by patients triaged to critical care $cca_j(t)$ and new ambulance arrivals $nab_j(t)$, and decreases as patients start consultation $cs_j(t)$. Patient consultation $cs_j(t)$ is initiated when an ED doctor becomes available and initiates consultation $nc_j(t)$. The equation for patients waiting for consultation $C_j(t)$ is:

$$C_j(t) = \int_{t_0}^{t} [cca_j(t) + nab_j(t) - cs_j(t)]dt + C_j(t_0) \qquad (11)$$

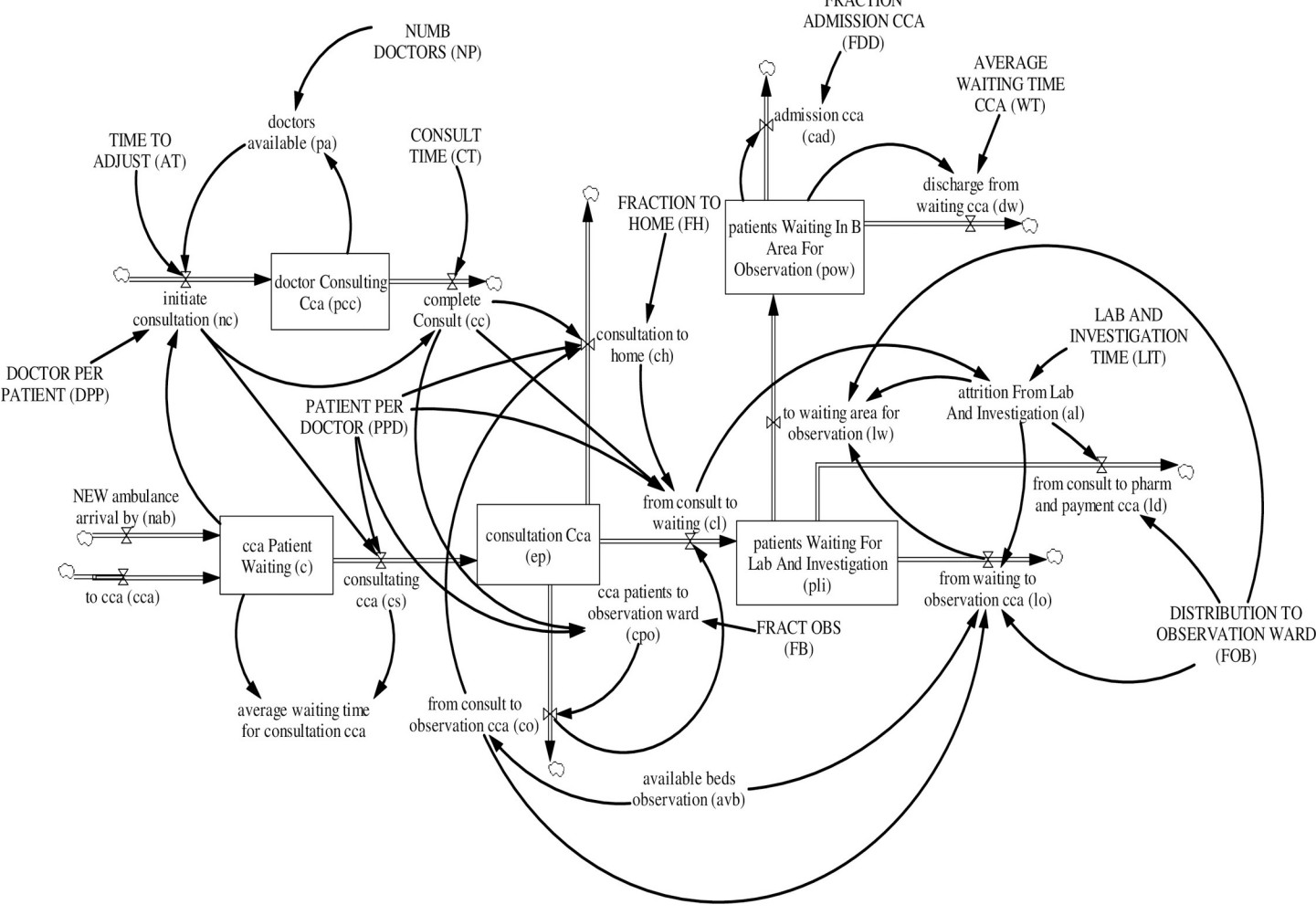

**Fig 6. Critical care area sub-model.**

where $C_j(t_0)$ is patients waiting for consultation at time $(t_0)$ and

$$nab_j(t) = exogenous\ data \tag{12}$$

$$cs_j(t) = nc_j(t) * ppd \tag{13}$$

$nab_j(t)$ is the exogenous historical ambulance arrival data; $ppd$ is the patient per doctor ratio in the critical care area.

Initiation of consultation requires an ED doctor. The number of ED doctors consulting $PCC(t)$ increases as ED doctors initiate consultation $nc_j(t)$ and decreases as consultation is completed $cc_j(t)$. ED doctors available to initiate consultation $pa(t)$ is the difference between the number of ED doctors allocated to critical care $NP(t)$ and ED doctors consulting $PCC(t)$. An average consultation time is assumed for each patient by treatment priority. P1 patients are assumed to require longer consultation time followed by P2, P3, and P4 patients. However, only P1 and P2 patients are triaged to the critical care area. The equation for the ED doctors

consulting $PCC(t)$ is:

$$PCC(t) = \int_{t_0}^{t} [nc_j(t) - cc_j(t)]dt + PCC(t_0) \tag{14}$$

where $PCC(t_0)$ is ED doctors consulting at the critical care area at time $(t_0)$ and

$$nc_{p1}(t) = MIN\left(pa(t), \frac{C_{p1}(t)}{AT}\right) \tag{15}$$

$$nc_{p2}(t) = MIN\left(\frac{pa(t)}{AT} - nc_{p1}(t), \frac{C_{p2}(t) * dpp}{AT}\right) \tag{16}$$

$$cc_j(t) = nc_j(t)(t - CT) \tag{17}$$

$$pa(t) = MAX(0, NP(t) - \sum PCC_j(t)) \tag{18}$$

$C_{p1}(t)$ and $C_{p2}(t)$ are P1 and P2 patients waiting for consultation in the critical care area; AT is adjustment time—a model artifact to ensure unit consistency. The value of AT is 1. CT is consultation time; $dpp$ is the doctor per patient ratio in the critical care area.

A co-flow structure was used to model patients in consultation. As an ED doctor initiates consultation $nc_j(t)$, a patient moves from the stock of patients waiting for consultation $C_j(t)$ to the stock of patients in consultation $EP_j(t)$. Hence, completion of consultation $cc_j(t)$ decreases the number of patients in consultation $EP_j(t)$via to observation $co_j(t)$, to laboratory and investigation $cl_j(t)$ or to home $ch_j(t)$. The equation for patients in consultation $EP_j(t)$ is:

$$EP_j(t) = \int_{t_0}^{t} [cs_j(t) - co_j(t) - cl_j(t) - ch_j(t)]dt + EP_j(t_0) \tag{19}$$

where $EP_j(t_0)$ is ED patients in consultation at time $(t_0)$ and

$$co_{p1}(t) = MIN(avb(t), \ cpo_{p1}(t)) \tag{20}$$

$$co_{p2}(t) = MIN(avb(t) - cpo_{p1}(t), \ cpo_{p2}(t)) \tag{21}$$

$$cpo_j(t) = (cc_j(t) * ppd) * fb \tag{22}$$

$$cl_j(t) = (cc_j(t) * ppd) - co_j(t) - ch_j(t) \tag{23}$$

$$ch_j(t) = ((cc_j(t) * ppd) - co_j(t)) * fh \tag{24}$$

$avb(t)$ is available beds in the observation ward; $cpo_{p1}(t)$ and $cpo_{p2}(t)$ are P1 and P2 patients requiring referral to the observation ward, $co_j(t)$ is the patients from consultation to observation, $fb$ is the fraction of patients who require observation, $fh$ is the fraction of patients discharged home after consultation.

After consultation, patients are either referred to the observation ward $co_j(t)$, to laboratory and investigation $cl_j(t)$ or discharged home $ch_j(t)$ depending on their care needs The number of patients waiting for laboratory and investigation $PLI_j(t)$, i.e., patients who have to go through the laboratory investigation process and wait for their results, increases as patients are referred to laboratory and investigation $cl_j(t)$ and decreases as patients are either discharged after laboratory and investigation $ld_j(t)$, transferred to the observation ward $lo_j(t)$, or observed at the waiting area due to lack of beds in the observation ward $lw_j(t)$. Patients under

observation in the waiting area $POW_j(t)$ are discharged $dw_j(t)$ as their conditions improve or admitted to the hospital $cad(t)$. The equations for patients waiting for laboratory and investigation $PLI_j(t)$ and patients under observation in the waiting area $POW_j(t)$ are:

$$PLI_J(t) = \int_{t_0}^{t} [cl_j(t) - lo_j(t) - ld_j(t) - lw_j(t)]dt + PLI_j(t_0) \tag{25}$$

$$POW_j(t) = \int_{t_0}^{t} [lw_j(t) - dw_j(t) - cad_j(t)]dt + POW_j(t_0) \tag{26}$$

where $PLI_j(t_0)$ is patients waiting for laboratory and investigation at time $(t_0)$, $POW_j(t_0)$ is patients under observation in the waiting area at time $(t_0)$, and

$$lo_{p1}(t) = MIN(avb(t) - \sum co_j(t), al_{p1}(t) * fob_{p1}) \tag{27}$$

$$lo_{p2}(t) = MIN(avb(t) - \sum co_j(t) - lo_{p1}(t), \ al_{p2}(t) * fob_{p2}) \tag{28}$$

$$ld_j(t) = al_j(t) * (1 - fob_j) \tag{29}$$

$$lw_{p1}(t) = MAX(0, (al_{p1}(t) * fob_{p1} - lo_{p1}(t))) \tag{30}$$

$$lw_{p2}(t) = MAX(0, (al_{p2}(t) * fob_{p2} - lo_{p2}(t))) \tag{31}$$

$$al_j(t) = cl_j(t - LIT) \tag{32}$$

$$dw_j(t) = POW_j(t)/wt \tag{33}$$

$$cad_j(t) = POW_j(t) * fdd \tag{34}$$

$al_j(t)$ is patients who have completed laboratory and investigation; $fob_j$ is the fraction of patients who need to go to the observation ward after laboratory and investigation; $LIT$ is the average waiting time for laboratory and investigation, $fdd$ is the fraction of patients waiting in the waiting area admitted; and $wt$ is the average observation time.

**3.4.3. Ambulatory care pathways.** Patients triaged to ambulatory care, like other care pathways, wait in queue for consultation. In total, there are two main pathways for patients triaged to the ambulatory care area (see Fig 7):

(i)  Waiting for consultation → consultation → laboratory investigation → discharge

(ii) Waiting for consultation → consultation → laboratory investigation → observation → discharge

The number of patients waiting for consultation $CAB_j(t)$ increases by patients triaged to ambulatory care $ab_j(t)$ and decreases as consultation starts $csAB_j(t)$. Patient consultation is initiated when an ED doctor becomes available and starts consultation $ncAB_j(t)$. The number of ED doctors consulting $PCAB(t)$ increases as an ED doctor initiates consultation $ncAB_j(t)$ and decreases as consultation is completed $ccAB_j(t)$. Available ED doctors to initiate consultation $paAB(t)$ is the difference between ED physicians allocated to ambulatory care $NPAB(t)$ and the number of ED physician consulting $PCAB_j(t)$. The equations for ED patients waiting for

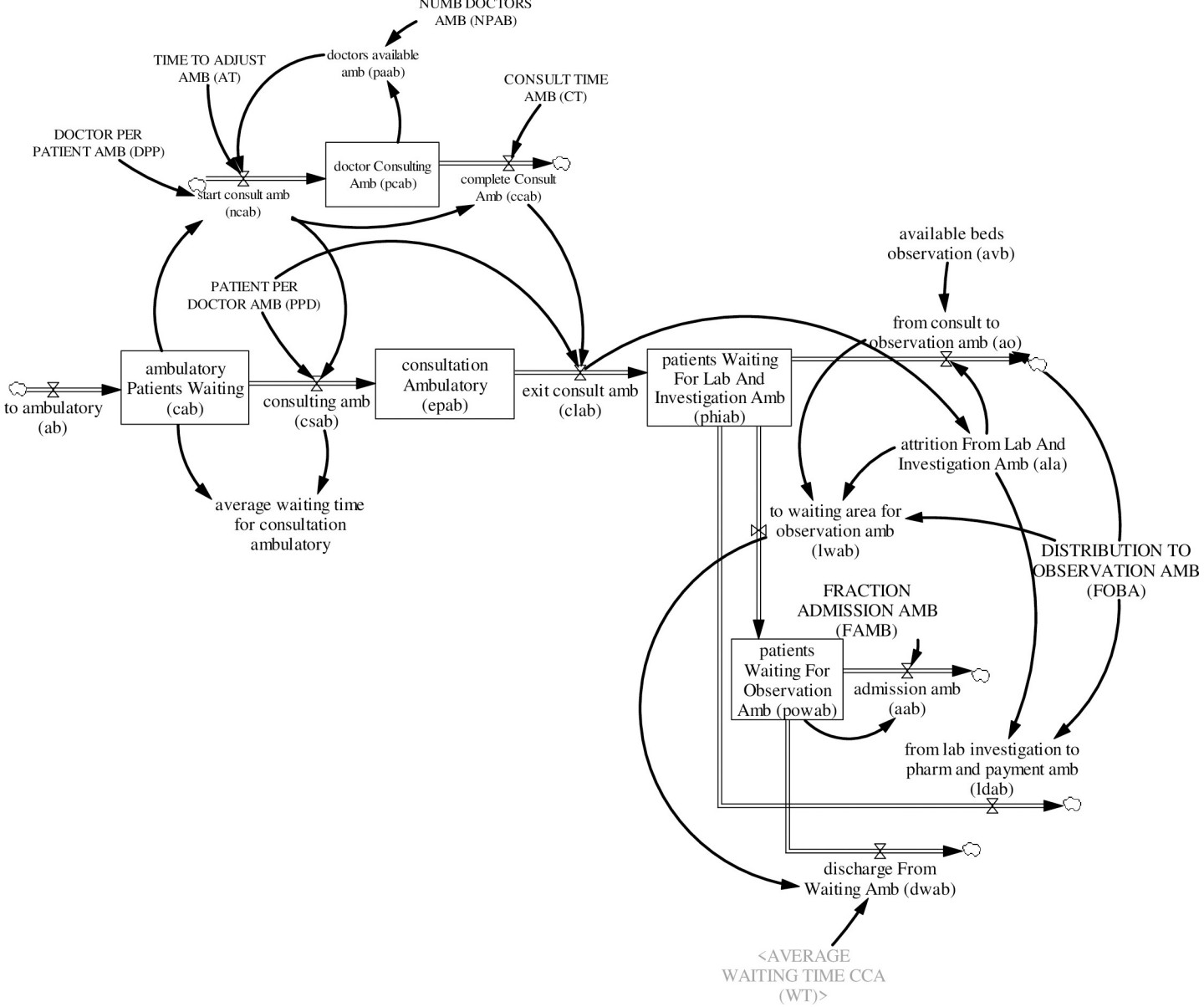

**Fig 7. Ambulatory care area sub-model.**

consultation $CAB_j(t)$ and ED physicians consulting $PCAB(t)$ are:

$$CAB_j(t) = \int_{t_0}^{t} [ab_j(t) - csAB_j(t)]dt + CAB_j(t_0) \tag{35}$$

$$PCAB(t) = \int_{t_0}^{t} [ncAB_j(t) - ccAB_j(t)]dt + PCAB(t_0) \tag{36}$$

where $CAB_j(t_0)$ is the number of patients in ambulatory care waiting for consultation at time $(t_0)$, $PCAB(t_0)$ is the number of ED physicians consulting at time $(t_0)$, and

$$csAB_j(t) = ncAB_j(t) * ppd \tag{37}$$

$$ncAB_{p1}(t) = MIN\left(paAB(t), \frac{CAB_{p1}(t)}{AT}\right) \tag{38}$$

$$ncAB_{p2}(t) = MIN\left(\frac{paAB(t)}{AT} - ncAB_{P1}(t), \frac{CAB_{p2}(t) * dpp}{AT}\right) \tag{39}$$

$$ncAB_{p3}(t) = MIN\left(\frac{paAB(t)}{AT} - ncAB_{P1}(t) - ncAB_{P2}(t), \frac{CAB_{p3}(t) * dpp}{AT}\right) \tag{40}$$

$$ncAB_{p4}(t) = MIN\left(\frac{paAB(t)}{AT} - ncAB_{P1}(t) - ncAB_{P2}(t) - ncAB_{P3}(t), \frac{CAB_{p4}(t) * dpp}{AT}\right) \tag{41}$$

$$ccAB_j(t) = ncAB_j(t)(t - CT) \tag{42}$$

$$paAB(t) = MAX(0, NPAB(t) - \sum PCAB_j(t)) \tag{43}$$

$ppd$ is the patient per doctor ratio in the ambulatory care area; $dpp$ is the doctor per patient ratio; $CT$ is consultation time, and AT is adjustment time.

Similar to the critical care area, a co-flow structure was developed to track patients in consultation in ambulatory care. As an ED physician initiates consultation $ncAB_j(t)$, a patient moves from the stock of patients waiting for consultation $CAB_j(t)$ to the stock of patients in consultation $EPAB_j(t)$. Consequently, completion of consultation $ccAB_j(t)$ decreases the stock of patients in consultation $EPAB_j(t)$ via to laboratory and investigation $clAB_j(t)$. The equation illustrating this dynamic is:

$$EPAB_j(t) = \int_{t_0}^{t}[csAB_j(t) - clAB_j(t)]dt + EPAB_j(t_0) \tag{44}$$

where $EPAB_j(t_0)$ is ED patients in consultation at the ambulatory care area at time $(t_0)$ and

$$clAB_j(t) = (ccAB_j(t) * ppd) \tag{45}$$

After consultation, patients proceed to laboratory and investigation. Patients waiting for laboratory and investigation $PHIAB_j(t)$–a procedure which includes various tests and examinations as well as waiting for test results to be discussed with the ED physician–increases as patients are referred to laboratory and investigation $clAB_j(t)$ and decreases as patients are referred to the observation ward $ao_j(t)$, discharged home $ldAB_j(t)$ or transferred to be observed in the waiting area due to limited beds in the observation ward $lwAB_j(t)$. Patients under observation in the waiting area $POWAB_j(t)$ due to capacity constraints in the observation ward decreases via discharge $dwAB(t)$ or hospital admission $aAB_j(t)$. The equations for patients in laboratory and investigation $PHIAB_j(t)$ and patients under observation in the waiting area $POWAB_j(t)$ are:

$$PHIAB_j(t) = \int_{t_0}^{t}[clAB_j(t) - ao_j(t) - lwAB_j(t) - ldAB_j(t)]dt + PHIAB_j(t_0) \tag{46}$$

$$POWAB_j(t) = \int_{t_0}^{t}[lwAB_j(t) - dwAB_j(t) - aAB_j(t)]dt + POWAB_j(t_0) \tag{47}$$

where $PHIAB_j(t_0)$ is the initial number of patients in the ambulatory care area waiting for laboratory and investigation at time $(t_0)$, $POWAB_j(t_0)$ is the number of patients under observation

at time ($t_0$), and

$$lwAB_{p1}(t) = MAX(0, (ala_{p1}(t) * foba_{p1} - ao_{p1}(t)))$$ (48)

$$lwAB_{p2}(t) = MAX(0, (ala_{p2}(t) * foba_{p2} - ao_{p2}(t)))$$ (49)

$$lwAB_{p3}(t) = MAX(0, (ala_{p3}(t) * foba_{p3} - ao_{p3}(t)))$$ (50)

$$lwAB_{p4}(t) = MAX(0, (ala_{p4}(t) * foba_{p4} - ao_{p4}(t)))$$ (51)

$$ldAB_j(t) = ala_j(t) * (1 - foba_j)$$ (52)

$$dwAB_j(t) = POWAB_j(t)/wt$$ (53)

$$aAB_j(t) = POWAB_j(t) * famb$$ (54)

$ala_j(t)$ is the number of patients who have finished laboratory and investigation, and $foba_j$ is the fraction of patients who have completed laboratory and investigation and are admitted to the observation ward, while $famb$ is the fraction of patients observed in the waiting area and are admitted to the hospital.

**3.4.4. Observation ward and discharge.**　The observation ward receives patients from critical care and ambulatory care areas—patients triaged to isolation care have a separate observation ward (see Fig 8). The number of patients in the observation ward $OW_j(t)$ increases as critical care patients are referred to it immediately after consultation $co_j(t)$ or after laboratory and investigation $lo_j(t)$, as well as referral of ambulatory patients after laboratory and investigation $ao_j(t)$, and decreases as patients are admitted into the hospital $ah_j(t)$ or discharged home via pharmacy and payment $do_j(t)$. Admission into the observation ward depends on the available beds $avb(t)$. Available beds $avb(t)$ is the difference between observation bed capacity $bc(t)$ and the number of patients in the observation ward $OW_j(t)$. After observation, discharged patients go through pharmacy and payment $PHAB_j(t)$ for payment and collection of prescribed medication. The number of patients in pharmacy and payment $PHAB_j(t)$ increases as patients are discharged from the observation ward $do_j(t)$, as patients are released from laboratory and investigations both in critical care $ld_j(t)$ and ambulatory care $ldAB_j(t)$, as well as patients are discharged from observation in waiting areas, both in critical care $dw_j(t)$ and ambulatory care $dwAB_j(t)$, as well as patients discharged after consultation from critical care $ch_j(t)$ and decreases as patients leave for home $hab_j(t)$. The equations for observation ward admission and discharge are:

$$OW_j(t) = \int_{t_0}^{t} [co_j(t) + lo_j(t) + ao_j(t) - ah_j(t) - do_j(t)]dt + OW_j(t_0)$$ (55)

$$PHAB_j(t) = \int_{t_0}^{t} [ldAB_j(t) + dwAB_j(t) + do_j(t) + ld_j(t) + dw_j(t) + ch_j(t) - hab_j(t)]dt + PHAB_j(t_0)$$ (56)

where $OW_j(t_0)$ is the initial number of patients in the observation ward at time ($t_0$), $PHAB_j(t_0)$

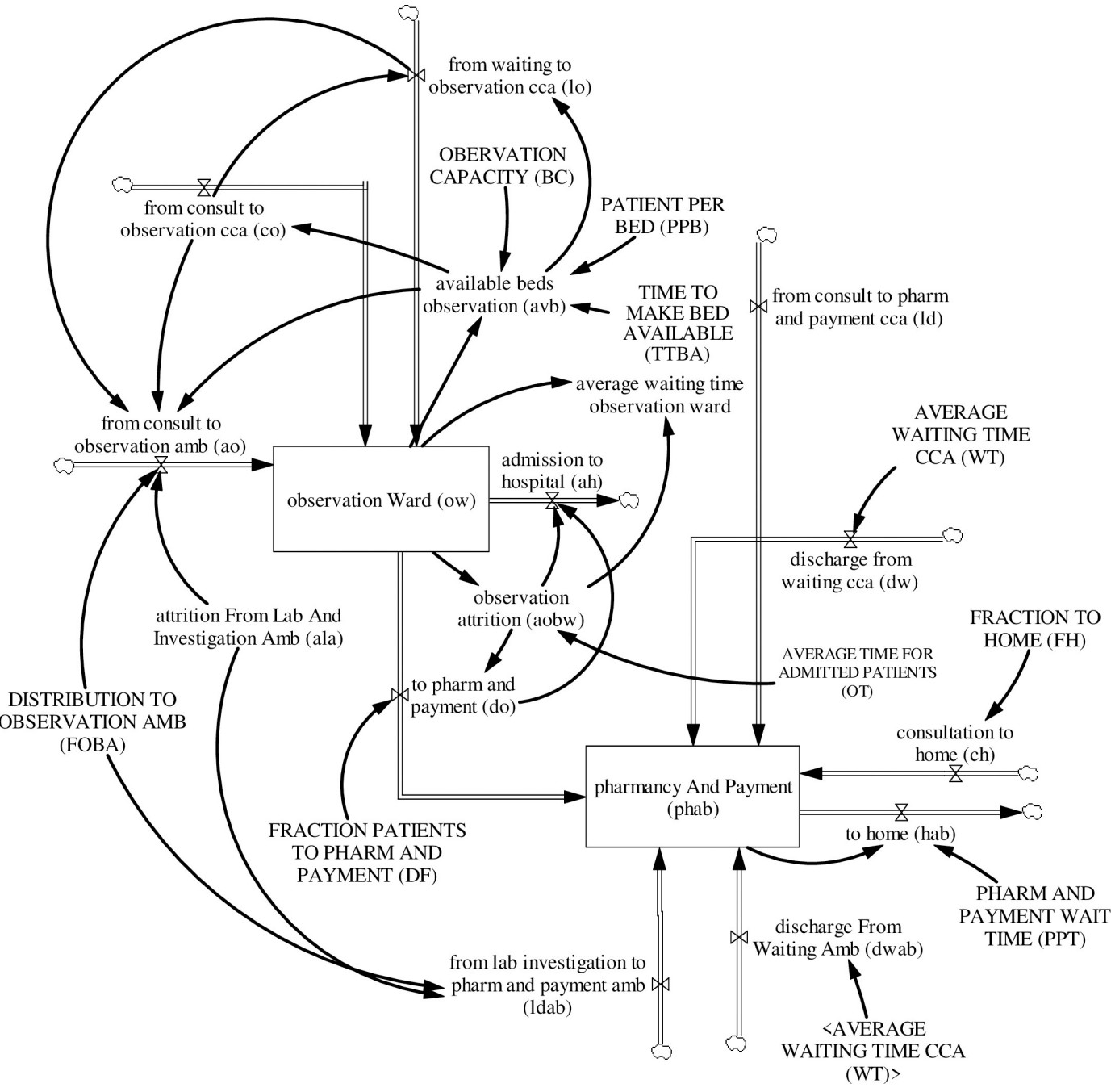

**Fig 8. Observation ward and discharge area sub-model.**

is the initial number of patients in the stock pharmacy and payment at time ($t_0$), and

$$ao_{p1}(t) = MIN(avb(t) - \sum co_j(t) - \sum lo_j(t), \ ala_{p1}(t) * foba_{p1}) \tag{57}$$

$$ao_{p2}(t) = MIN(avb(t) - \sum co_j(t) - \sum lo_j(t) - ao_{p1}(t), ala_{p2}(t) * foba_{p2}) \tag{58}$$

$$ao_{p3}(t) = MIN(avb(t) - \sum co_j(t) - \sum lo_j(t) - ao_{p1}(t) - ao_{p2}(t), ala_{p3}(t) * foba_{p3}) \quad (59)$$

$$ao_{p4}(t) = MIN(avb(t) - \sum co_j(t) - \sum lo_j(t) - ao_{p1}(t) - ao_{p2}(t) - ao_{p3}(t), ala_{p4}(t)$$
$$* foba_{p4}) \quad (60)$$

$$ah_j(t) = aobw_j(t) - do_j(t) \quad (61)$$

$$aobw_j(t) = OW_t/ot \quad (62)$$

$$do_j(t) = aobw_j(t) * df \quad (63)$$

$$avb(t) = ((bc(t) * ppb) - \sum ow_j(t))/ttba \quad (64)$$

where $avb(t)$ is the total available beds in the observation ward; $ala_j(t)$ is the number of patients who have finished laboratory and investigation; $foba_j$ is the fraction of patients who have completed laboratory and investigation and are admitted to the observation ward; $df$ is the fraction of discharged patients from the observation ward; $ot$ is the average observation time for patients in the observation ward; $ppb$ is the patients per bed ratio; $ttba$ is the time to make a bed available for patients to use.

**3.4.5. Isolation care pathways.** Patients triaged to isolation care, like other care pathways, wait in queue for consultation. In total, there are two main pathways for patients triaged to isolation care (see Fig 9):

(i)  Waiting for consultation → consultation → discharge

(ii)  Waiting for consultation → consultation → observation → discharge

The number of patients waiting for consultation $CIS_j(t)$ increases by patients triaged to isolation care $is_j(t)$ and decreases as consultation starts $csIS_j(t)$. Patient consultation is initiated when an ED physician becomes available and initiates consultation $ncIS_j(t)$. The number of ED physicians consulting $PCIS(t)$ at the isolation care area increases as an ED physician starts consultation $ncIS_j(t)$ and decreases as consultation is completed $ccIS_j(t)$. Available ED physicians to initiate consultation $paIS(t)$ is the difference between ED physicians allocated to isolation care $NPIS(t)$ and ED physicians currently consulting with patients $PCIS_j(t)$. The equations for patients waiting for consultation $CIS_j(t)$ and ED doctors consulting in the isolated care area $PCIS(t)$ are:

$$CIS_j(t) = \int_{t_0}^{t} [is_j(t) - csIS_j(t)]dt + CIS_j(t_0) \quad (65)$$

$$PCIS(t) = \int_{t_0}^{t} [ncIS_j(t) - ccIS_j(t)]dt + PCIS(t_0) \quad (66)$$

where $CIS_j(t_0)$ is patients in the isolation care area waiting for consultation at time $(t_0)$, $PCIS$

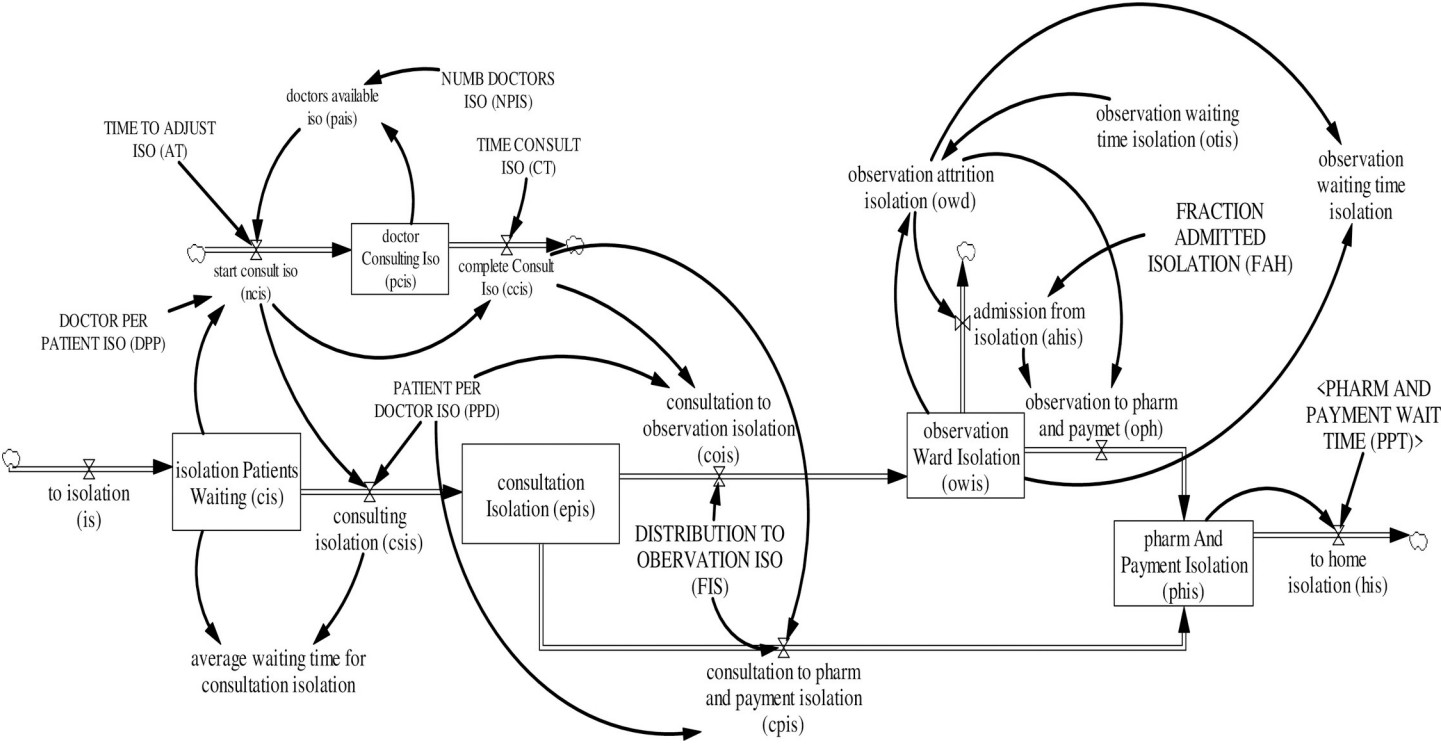

**Fig 9. Isolation area sub-model.**

$(t_0)$ is the number of ED physicians consulting at time $(t_0)$, and

$$csIS_j(t) = ncIS_j(t) * ppd \tag{67}$$

$$ncIS_{p1}(t) = MIN\left(paIS(t), \frac{CIS_{p1}(t)}{AT}\right) \tag{68}$$

$$ncIS_{p2}(t) = MIN\left(\frac{paIS(t)}{AT} - ncIS_{P1}(t), \frac{CIS_{p2}(t) * dpp}{AT}\right) \tag{69}$$

$$ncIS_{p3}(t) = MIN\left(\frac{paIS(t)}{AT} - ncIS_{P1}(t) - ncIS_{P2}(t), \frac{CIS_{p3}(t) * dpp}{AT}\right) \tag{70}$$

$$ncIS_{p4}(t) = MIN\left(\frac{paIS(t)}{AT} - ncIS_{P1}(t) - ncIS_{P2}(t) - ncIS_{P3}(t), \frac{CIS_{p4}(t) * dpp}{AT}\right) \tag{71}$$

$$ccIS_j(t) = ncis(t)(t - CT) \tag{72}$$

$$paIS(t) = MAX(0, NPIS(t) - \sum PCIS_j(t)) \tag{73}$$

where $ppd$ is the patient per doctor ratio in the isolation care area; $dpp$ is the doctor per patient ratio; AT is adjustment time; and $CIS_{P1}$, $CIS_{p2}$, $CIS_{P3}$, and $CIS_{P4}$ are the stocks of patients waiting for consultation.

Similar to other care areas, a co-flow structure was developed to model patients in consultation in the isolation care area. As an ED physician initiates consultation $ncIS_j(t)$ a patient moves from the stock of patients waiting for consultation $csIS_j(t)$ to the stock of patients in consultation $EPIS_j(t)$. Hence, completion of consultation $ccIS_j(t)$ decreases the stock of patients in consultation via to observation $coIS_j(t)$ and for discharge $cpIS_j(t)$. After consultation, patients referred to the observation ward $coIS_j(t)$ are observed in the observation ward $OWIS_j(t)$. After a period of observation time $otis$, patients in the observation ward are either discharged $oph_j(t)$ or admitted into the hospital $ahis_j(t)$. Likewise, patients discharged from the isolation care area, i.e., via observation ward $oph_j(t)$ and or after consultation $cpIS_j(t)$, proceed to the pharmacy and payment $PHIS_j(t)$ for prescribed medicine and payment and then leave $his_j(t)$. The equations illustrating the stock of patients in consultation $EPIS_j(t)$, the stock of patients in observation $OWIS_j(t)$, and the stock of patients in pharmacy and payment $PHIS_j(t)$ are:

$$EPIS_j(t) = \int_{t_0}^{t} [csIS_j(t) - coIS_j(t) - cpIS_j(t)]dt + EPIS_j(t_0) \tag{74}$$

$$OWIS_j(t) = \int_{t_0}^{t} [coIS_j(t) - oph_j(t) - ahis_j(t)]dt + OWIS_j(t_0) \tag{75}$$

$$PHIS_j(t) = \int_{t_0}^{t} [oph_j(t) + cpIS_j(t) - his_j(t)]dt + PHIS_j(t_0) \tag{76}$$

where $EPIS_j(t_0)$ is the initial number of patients in consultation at the isolation care area at time $(t_0)$, $OWIS_j(t_0)$ is the number of patients under observation at time $(t_0)$, $PHIS_j(t_0)$ is the number of patients in pharmacy and payment at time $(t_0)$, and

$$coIS_j(t) = (ccIS_j(t) * ppd) * fis \tag{77}$$

$$cpIS_j(t) = (ccIS_j(t) * ppd) * (1 - fis) \tag{78}$$

$$ahis_j(t) = owd_j(t) * fah \tag{79}$$

$$oph_j(t) = owd_j(t) - ahis_j(t) \tag{80}$$

$$owd_j(t) = owis_j(t)/otis \tag{81}$$

$$his_j(t) = phis(t)/ppt \tag{82}$$

where $fis$ is the fraction of patients who need to go to the observation ward; $owd_j(t)$ is the number of patients who were observed and proceed to discharge or hospital admission; $fah$ is the fraction of patients who were observed and are admitted to the hospital; $otis$ is the average time patients were observed; $ppt$ is the average time at the pharmacy and payment.

### 3.5. Data sources

We parameterized a simulation model that runs for 24 hours. To that end, we had access to ED data for the period of June–August 2017. The key parameter values essential for estimating ALOS are *average registration time*, *average triage time*, *consult time* (for critical care, ambulatory care and isolation care), *average waiting time for observation ward*, *lab and investigation waiting time*, *pharmacy and payment waiting time*, *time to make bed available*, and *average time to admit patients*. The distribution of the key parameter values is assumed to follow a

triangular distribution [38]. The estimated values used for the lower limit *a*, upper limit *b*, and mode *c* are as follows: For registration and triage, the values are—*average registration time* [Min = 3; Median = 5; Max = 7], and *average triage time* [Min = 4; Median = 5; Max = 7]. For critical care the input values are—*consult time* [Min = 10; Median = 15; Max = 20], *average observation waiting time* [Min = 30; Median = 60; Max = 90], and *laboratory and investigation* [Min = 35; Median = 45; Max = 60]. For ambulatory care, the input values are—*consult time* [Min = 10; Median = 15.5; Max = 20], and *average observation waiting time* [Min = 30; Median = 60; Max = 90]. For isolation care, the values are—*consult time* [Min = 20; Median = 30; Max = 45], *average observation waiting time* [Min = 30; Median = 60; Max = 90], and *pharmacy and payment waiting time* [Min = 10; Median = 15; Max = 30]. For observation ward and discharge, the input values are—*time to make beds available* [Min = 10; Median = 15; Max = 25], and *average time to admit patients* [Min = 100; Median = 120; Max = 150].

Referring to patient arrivals, we picked the highest daily patient arrival pattern in the 3-months period from June to August 2017 because we intentionally wanted to stress-test the system. Operational problems in the ED become only visible when the workload is high, and the system is stretched to its limits.

To supplement the data requirements, an observational study on other process timings that were not captured in the patient records was conducted over a 2 weeks period in November 2017 to estimate some of the model parameters. Finally, for all parameters that could not be observed nor estimated from the data, we had to rely on expert judgment. More information about model inputs, their values, units, and sources can be found in S1 Table. The simulation model is provided in the supplementary file for review.

## 3.6. Model validation

To ensure that the ED model developed herein is fit for purpose and robust and that the results could be used to inform ED policies, structure and behavior validation tests were conducted [39–41]. Structure validation tests focused on engaging ED doctors with significant experience in the operations of the ED in Singapore to verify the model structure and its assumptions regarding causal relationships, feedbacks, time delays, and patient flows. This validation process was conducted in four different meetings—where the model structure was thoroughly reviewed—to ensure that the model structure is as close to reality as possible. Hence, we believe that the current model structure is firmly grounded in current operations of a hospital-based ED in Singapore.

On behavior validation tests, the simulation results were compared to available data of selected outcomes (average length of stay across all venues—CCA, ambulatory, and isolation). In addition, a mean absolute percentage error (MAPE) and a Theil statistic [42, 43] analysis were conducted to check the behavioral validity of the model. The MAPE—which is a measure of prediction accuracy—for the selected outcomes were; 9.78% for ALOS in the critical care area, 12.5% for ALOS in the ambulatory care area, and 8.85% for ALOS in the isolation care area. Given that MAPE of 30% is considered to be good, our results—which have a maximum MAPE of 12.5%—indicate that the simulated model compares well with available data considering that available ALOS data used was an average across venues over an hour. For the Theil statistic, the error due to bias ($U^M$) for CCA, ambulatory, and isolation care areas were 11%, 4.5%, and 17.7% respectively; while that for unequal variance ($U^S$) for CCA, ambulatory, and isolation care areas was 2.4%, 7.1%, and 9.4% respectively; and the error for covariation component ($U^C$) for CCA, ambulatory, and isolation care areas were 87.1%, 88.4%, and 72.9% respectively. Thus, critical care, ambulatory care, and isolation care areas have most of the error within the covariation component ($U^C$) as compared to bias ($U^M$) and unequal variance

($U^S$). For Theil statistics, if majority of the errors comes from covariation components, it indicates that the simulated variables track the underlying trend well, but diverge when comparing point-by-point, indicating that majority of the errors are unsystematic with respect to the purpose of the model.

## 4. Policy experiments

Policy experimentation was conducted to explore the range of potential future directions on how to manage ED crowding as discussed with the ED physicians, as well as address the identified bottlenecks—significant waiting times for consultation, laboratory investigation and observation ward for admission—in the ED care processes. To that end, we tested the impact of four policies on the average length of stay (ALOS) of ED patients by venue of care, care pathway, and time period of the day. Policies were compared to the base case, where the status quo is simulated. We use ALOS as a proxy for ED crowding [44–46]. The tested policies are:

1. **Business-as-usual (BAU)**: The business-as-usual (BAU) or base-case experiment assumes no change to key model inputs that may be affected by current or future policies. Under this policy experiment, patients reporting at the ED for care are triaged to CCA, ambulatory care, or isolation care. The current allocation of ED doctors across the venues remains unchanged and waiting time for patients in the observation ward is assumed to remain constant. This hypothetical scenario is unlikely in the current context as new policies are expected to change some of these key variables. However, it is included to serve as a reference point for evaluating alternative policies.

2. **Co-location** (policy 1): This policy experiment varies the fraction of P4 and P3 patients decanted from the ED to a GP clinic co-located in the ED from 10% to 30% to assess the impact on ALOS. The rationale of this policy is to relieve ED Operations by redirecting non-emergency patients to primary care services. In Singapore and other industrialized countries, there is a tendency to co-locate primary care services within EDs. These primary care services are clearly separated from ED operations where more severely ill patients are treated. In such a way, the flow of patients with only minor and non-emergency symptoms are cared for in a separate venue with own resources. This reduces the heterogeneity in acuity of patients in the other venues of the ED. Overall, this policy aims to increase efficiency and effectiveness of ED operations by diverting non-emergency patients to GP care.

3. **Capacity of doctors** (policy 2): This policy experiment stepwise increases the capacity of ED doctors by 10%, 20%, and 30%, across all venues of care, to evaluate its impact on waiting time and ALOS. This policy experiment aims to achieve the target ALOS of 4 hours for patients allocated to critical care and ambulatory care and 4.5 hours for patients triaged to isolation care. Singapore loosely follows the '4-hour rule' from the UK where 98% of all ED patients must be seen and discharged or admitted within 4 hours of their arrival. In Singapore and other wealthy countries such as Switzerland, managers of EDs typically tried to meet waiting time targets by hiring more physicians and nurses. So, this policy simply reflects the attempt to balance the increased demand for emergency medical services by increasing the supply of these services. However, there is a clear financial limit to such a policy. As more and more health systems are forced to reduce spending and become more efficient, a policy of matching increased demand with increased supply might not be a viable one in the long-term.

4. **Observation ward and laboratory** (policy 3): This policy explores the impact of a 10%-30% reduction in waiting time at the observation ward, as well as waiting time for laboratory

and investigation on ALOS across all venues of care. This policy experiment aims to achieve the target ALOS of 4 hours for patients allocated to critical care and ambulatory care and 4.5 hours for patients triaged to isolation care. The rationale of this policy is to analyze the consequences of an improved patient outflow from the ED. One of the most severe bottlenecks of a hospital-based ED is the outflow of patients who cannot be discharged but need to be admitted to the hospital. Because of limited hospital capacities these patients typically accumulate in the ED and gradually fill up the observation ward. Consequently, these 'boarded' patients block ED resources and create significant inefficiencies in the system. This policy analyzes the implications of an improved transfer of ED patients to hospital wards by reducing the waiting time at the observation ward (i.e., the boarding time). Furthermore, this policy tests the consequences of an increased turnover for laboratory and investigation (e.g., more efficient blood testing).

5. **Combined interventions** (policy 4): This policy experiment implements all the previous interventions—i.e., co-location, capacity of doctors, and observation ward and laboratory—simultaneously to assess it impact on ALOS across all venues of care.

## 5. Results

S1–S4 Figs show the ALOS by venue of care—critical care, ambulatory care, and isolation care—care pathway and time of the day, for all the policy experiments. To present the result in a meaningful way (as shown in Tables 1–4), we divided the day into three time periods—herein referred to as phases. *Phase 1* (ph1) is from 00:00 am to 08:00 am, *phase 2* (ph2) is from 08:00 am to 04:00 pm, and *phase 3* (ph3) is from 04:00 pm to 00:00 am. The results are presented as follows:

### 5.1. Business-as-usual (BAU)

As shown in Tables 1–4, in the BAU case, a critical care patient that goes through critical care pathway 1 for emergency care is projected to experience an ALOS of **25 minutes** from *00:00 am to 08:00 am*; **39 minutes** from *08:00 am to 04:00 pm*; and **27 minutes** from *04:00 pm to 00:00 am*. The estimated ALOS for critical care pathway 2 patients is **72 minutes** from *00:00 am to 08:00 am*; **85 minutes** from *08:00 am to 04:00 pm*; and **73 minutes** from *04:00 pm to*

**Table 1. ALOS ED patients face depending on care venue, patient pathway, arrival time for co-location policy where 10% - 30% of P4 and P3 patients are decanted from the ED to a GP clinic co-located in the ED.**

| Policy 1 | Critical care pathways (minutes) | | | | | | | | | | | |
| | Pathway 1 | | | Pathway 2 | | | Pathway 3 | | | Pathway 4 | | |
| | Ph1 | Ph2 | Ph3 | Ph1 | Ph2 | Ph3 | Ph1 | Ph2 | Ph3 | Ph1 | Ph2 | Ph3 |
| BAU | 25 | 39 | 27 | 72 | 85 | 73 | 149 | 162 | 150 | 195 | 209 | 197 |
| 10% | 25 | 39 | 27 | 72 | 85 | 73 | 149 | 162 | 150 | 195 | 209 | 197 |
| 20% | 25 | 39 | 27 | 72 | 85 | 73 | 149 | 162 | 150 | 195 | 209 | 197 |
| 30% | 25 | 39 | 27 | 72 | 85 | 73 | 149 | 162 | 150 | 195 | 209 | 197 |
| | Ambulatory care pathways (minutes) | | | | | | Isolation care pathways (minutes) | | | | | |
| | Pathway 1 | | | Pathway 2 | | | Pathway 1 | | | Pathway 2 | | |
| | Ph1 | Ph2 | Ph3 | Ph1 | Ph2 | Ph3 | Ph1 | Ph2 | Ph3 | Ph1 | Ph2 | Ph3 |
| BAU | 72 | 166 | 85 | 196 | 290 | 208 | 42 | 42 | 42 | 101 | 101 | 102 |
| 10% | 72 | 125 | 81 | 196 | 249 | 205 | 42 | 42 | 42 | 101 | 101 | 102 |
| 20% | 72 | 106 | 78 | 196 | 230 | 201 | 42 | 42 | 42 | 101 | 101 | 102 |
| 30% | 72 | 96 | 75 | 196 | 219 | 199 | 42 | 42 | 42 | 101 | 101 | 102 |

*00:00 am*. The ALOS for critical care pathway 3 patients is **149 minutes** from *00:00 am to 08:00 am*, **162 minutes** from *08:00 am to 04:00 pm* and **150 minutes** from *04:00 pm to 00:00 am*. Lastly, the ALOS for critical care pathway 4 patients is **195 minutes** from *00:00 am to 08:00 am*, 209 minutes from *08:00 am to 04:00 pm* and 197 minutes from *04:00 pm to 00:00 am*.

For patients seeking emergency care at the ambulatory care venue, and who go through ambulatory care pathway 1 are estimated to experience an ALOS of **72 minutes** from *00:00 am to 08:00 am*; **166 minutes** from *08:00 am to 04:00 pm*; and **85 minutes** from *04:00 pm to 00:00 am*. For ambulatory pathway 2 patients, under the base-case, a projected ALOS of **196 minutes** from *00:00 am to 08:00 am*; **290 minutes** from *08:00 am to 04:00 pm*; and **208 minutes** from *04:00 pm to 00:00 am* is expected.

Lastly, patients at the isolation care area following the isolation care pathway 1 are projected to experience an ALOS of **42 minutes** at all phases. For isolation care pathway 2 patients, ALOS is projected to be **101 minutes** from *00:00 am to 04:00 pm* and **102 minutes** from *04:00 pm to 00:00 am*.

## 5.2. Co-location (policy 1)

As indicated in Table 1, under policy 1, where 10% to 30% of P4 and P3 patients from each venue of care are decanted from the ED to a GP clinic co-located in the ED to provide needed care, the ALOS for critical care patients going through critical care pathways 1 to 4 are projected to be the same as BAU. However, for patients going through ambulatory care pathway 1 and 2, ALOS was projected to reduce under policy 1. In the scenario where 10% of P4 and P3 patients were decanted, ALOS for ambulatory care pathway 1 is projected to decrease by **24.6%** from *08:00 am to 04:00 pm*, and by **4.7%** from *04:00 pm to 00:00 am*, compared to the BAU. The ALOS of patients in ambulatory care pathway 1 arriving in the time period between *00:00 am and 08:00 am* remains unchanged for the 10%, 20%, and 30% scenario; for ambulatory care pathway 2 ALOS is projected to decrease by **14.1%** from *08:00 am to 04:00 pm*, while that for *04:00 pm to 00:00* am is **1.44%**, compared to the BAU. The ALOS of patients in ambulatory care pathway 2 presenting themselves between *00:00 am and 08:00 am* remains unchanged for the 10%, 20%, and 30% scenario. In the scenario where 20% of P4 and P3 patients were removed from the ED, ALOS for ambulatory care pathway 1 is projected to fall by **36.1%** from *08:00 am to 04:00 pm*, while that for *04:00 pm to 00:00 am* is **8.2%**, compared to the BAU; for ambulatory care pathway 2 ALOS is forecasted to diminish by **20.7%** from *08:00 am to 04:00 pm*, while that for *04:00 pm to 00:00 am* is **3.4%**, compared to the BAU. Finally, in the scenario where 30% of P4 and P3 patients are redirected to an inhouse GP clinic, ALOS for ambulatory care pathway 1 is projected to fall by **42.2%** from *08:00 am to 04:00 pm* and by **11.8%** from *04:00 pm to 00:00 am*, compared to the BAU; for ambulatory care pathway 2 ALOS is forecasted to reduce by **24.5%** from *08:00 am to 04:00 pm* and by **4.3%** from *04:00 pm to 00:00 am*, compared to the base case. Lastly, policy 1 does not show an impact on isolation care pathways in our experimental set-up. On average, only 6% of all presenting patients are triaged to isolation care (i.e., fever patients) and so patient flow is smooth through this care venue. This means that fever patients typically do not need to wait until they see a doctor. Consequently, reducing the inflow of patients into isolation care (which is the effect of policy 1) does not change the ALOS of patients going through this venue of care. ALOS is driven by the waiting time for lab and investigation and if necessary, by the waiting time for a bed in the hospital. Both waiting times are not influenced by policy 1.

**Table 2. ALOS ED patients face depending on care venue, patient pathway, arrival time for policy where the numbers of doctors are increased from 10% - 30% across all venues of care.**

| Policy 2 | Critical care pathways (minutes) | | | | | | | | | | | |
|---|---|---|---|---|---|---|---|---|---|---|---|---|
| | Pathway 1 | | | Pathway 2 | | | Pathway 3 | | | Pathway 4 | | |
| | Ph1 | Ph2 | Ph3 | Ph1 | Ph2 | Ph3 | Ph1 | Ph2 | Ph3 | Ph1 | Ph2 | Ph3 |
| BAU | 25 | 39 | 27 | 72 | 85 | 73 | 149 | 162 | 150 | 195 | 209 | 197 |
| 10% | 26 | 31 | 26 | 72 | 78 | 73 | 149 | 155 | 149 | 195 | 201 | 196 |
| 20% | 26 | 29 | 26 | 72 | 75 | 72 | 149 | 152 | 149 | 195 | 199 | 196 |
| 30% | 25 | 28 | 26 | 72 | 74 | 72 | 149 | 151 | 149 | 195 | 198 | 195 |
| | Ambulatory care pathways (minutes) | | | | | | Isolation care pathways (minutes) | | | | | |
| | Pathway 1 | | | Pathway 2 | | | Pathway 1 | | | Pathway 2 | | |
| | Ph1 | Ph2 | Ph3 | Ph1 | Ph2 | Ph3 | Ph1 | Ph2 | Ph3 | Ph1 | Ph2 | Ph3 |
| BAU | 72 | 166 | 85 | 196 | 290 | 208 | 42 | 42 | 42 | 101 | 101 | 102 |
| 10% | 72 | 109 | 79 | 196 | 232 | 203 | 42 | 42 | 42 | 101 | 101 | 102 |
| 20% | 72 | 92 | 76 | 196 | 216 | 199 | 42 | 42 | 42 | 101 | 101 | 102 |
| 30% | 72 | 84 | 74 | 196 | 208 | 197 | 42 | 42 | 42 | 101 | 101 | 102 |

### 5.3. Capacity of doctors (policy 2)

As shown in Table 2, under this policy where the number of ED doctors is gradually increased by 10%, 20% and 30%, a critical care patient going through critical care pathways 1 to 4 is projected to experience ALOS similar to that of the BAU from *00:00 am to 08:00 am* and *04:00 pm to 00:00 am*. However, during the time period of *08:00 am to 04:00 pm*, with a scenario where the doctors allocated to each venue is increased by 10%, ALOS is projected to decrease by **20.5%**, **8.2%**, **4.3%** and **3.8%** respectively for care pathways 1 to 4. In the scenarios where a 30% increase of doctor's allocation was experimented, ALOS is projected to decrease by **28.2%**, **12.9%**, **6.7%** and **5.2%** respectively for care pathways 1 to 4.

Similarly, under the ambulatory care pathways 1 and 2, ALOS is projected to be similar to that of the BAU from *00:00 am to 08:00 am*. However, ALOS is projected to decrease under the 10% increase in doctor's capacity scenario by **34.3%**, and **20%** for ambulatory care pathways 1 and 2 respectively from *08:00 am to 04:00 pm*; while that for *04:00 pm to 00:00 am* was **7.05%** and **2.4%**. Under the 30% increase in doctor's capacity scenario, ALOS is expected to reduce by **49.3%** from *08:00 am to 04:00 pm* and **12.9%** from *04:00 pm to 00:00 am* for ambulatory care pathway 1; and **28.2%** from *08:00 am to 04:00 pm* and **5.2%** from *04:00 pm to 00:00 am* for ambulatory care pathway 2. Lastly, ALOS for patients going through isolation care pathways are expected to experience ALOS like that of the BAU. The drivers of ALOS for patients in isolation care are the waiting times for lab and investigation and admission to the hospital. Both waiting times are not changed (reduced) by increasing the capacity of doctors in the ED.

### 5.4. Observation ward and laboratory (policy 3)

As shown in Table 3, under the observation ward and laboratory policy, a critical care patient going through critical care pathways 1 is projected to experience ALOS comparable to that of the BAU across all time phases. But, for critical care pathway 2, 3 and 4, ALOS is projected to decrease as observation ward and laboratory and investigation waiting times are reduced. Under the scenario where observation ward and laboratory waiting times were reduced by 10%, compared to the BAU, ALOS is expected to decrease by **6.9%**, **4.7%** and **5.4%** for critical care pathway 2 across the three-time phases; **8%**, **7.4%** and **8%** for critical care pathway 3

**Table 3. ALOS ED patients face depending on care venue, patient pathway, arrival time for observation ward and laboratory policy where waiting times at the observation ward, and laboratory are reduced by 10% - 30%.**

| Policy 3 | Critical care pathways (minutes) | | | | | | | | | | | |
|---|---|---|---|---|---|---|---|---|---|---|---|---|
| | Pathway 1 | | | Pathway 2 | | | Pathway 3 | | | Pathway 4 | | |
| | Ph1 | Ph2 | Ph3 | Ph1 | Ph2 | Ph3 | Ph1 | Ph2 | Ph3 | Ph1 | Ph2 | Ph3 |
| BAU | 25 | 39 | 27 | 72 | 85 | 73 | 149 | 162 | 150 | 195 | 209 | 197 |
| 10% | 25 | 39 | 27 | 67 | 81 | 69 | 137 | 150 | 138 | 178 | 192 | 180 |
| 20% | 25 | 39 | 27 | 63 | 76 | 64 | 124 | 137 | 126 | 161 | 175 | 163 |
| 30% | 25 | 39 | 27 | 58 | 71 | 59 | 112 | 125 | 113 | 144 | 158 | 146 |
| | Ambulatory care pathways (minutes) | | | | | | Isolation care pathways (minutes) | | | | | |
| | Pathway 1 | | | Pathway 2 | | | Pathway 1 | | | Pathway 2 | | |
| | Ph1 | Ph2 | Ph3 | Ph1 | Ph2 | Ph3 | Ph1 | Ph2 | Ph3 | Ph1 | Ph2 | Ph3 |
| BAU | 72 | 166 | 85 | 196 | 290 | 208 | 42 | 42 | 42 | 101 | 101 | 102 |
| 10% | 67 | 161 | 80 | 179 | 273 | 191 | 42 | 42 | 42 | 95 | 95 | 96 |
| 20% | 63 | 157 | 76 | 162 | 256 | 174 | 42 | 42 | 42 | 89 | 89 | 90 |
| 30% | 58 | 152 | 71 | 145 | 239 | 157 | 42 | 42 | 42 | 84 | 83 | 84 |

across the three-time phases; while that for critical care pathway 4 was **8.7%**, **8.1%** and **8.6%** respectively across the three phases. Under the scenario where observation ward and laboratory waiting times were reduced by 30%, ALOS for critical care pathway 2 is projected to reduce by **19.4%** from *00:00 am to 08:00 am*, **16.4%** from *08:00 am to 04:00 pm* and **19.1%** from *04:00 pm to 00:00 am*; while that for critical care pathway 3 was **24.8%** from *00:00 am to 08:00 am*, **22.8%** from *08:00 am to 04:00 pm* and **24.6%** from *04:00 pm to 00:00 am*. Likewise, the ALOS for critical care pathway 4 is projected to reduce by **26.15%** from *00:00 am to 08:00 am*, **24.4%** from *08:00 am to 04:00 pm* and **25.8%** from *04:00 pm to 00:00 am*.

Considering the ambulatory care pathways, under the 10% reduction in observation ward and laboratory waiting times ALOS is projected to decrease **6.9%**, **3%** and **5.8%** respectively across the time phases for care pathway 1; while that for care pathway 2 is projected to decrease by **8.6%**, **5.8%** and **8.1%** respectively across the time phases. Under the 30% reduction in waiting times, ALOS is projected to decrease by **19.4%**, **8.4%** and **16.4%** respectively for care pathway 1, while projections for care pathway 2 are reductions of **26%**, **18.6%** and **24.5%** respectively.

For isolation care pathways, while isolation care pathway 1 is projected to remain unchanged relative to the BAU, isolation care pathway 2, under the 10% reduction in observation ward and laboratory waiting times is projected to decrease ALOS by **5.9%** across all time phases. However, under the 30% reduction in observation ward and laboratory waiting times, ALOS is projected to decrease by **16.8%**, **17.8%** and **17.6%** respectively across the time phases.

## 5.5. Combined interventions (policy 4)

As indicated in Table 4, under the combined interventions where policies 1 to 3 are implemented simultaneously, under the 10% assumptions—where 10% of P4 and P3 patients are decanted to a GP clinic in the ED, 10% increase in doctors allocated to each care venue and 10% reduction in observation ward and laboratory waiting times—a critical care patient that goes through critical care pathway 1 for emergency care is projected to experience **20.5%** reduction in ALOS from *08:00 am to 04:00 pm*; while that for *04:00 pm to 00:00 am* is **3.7%**. Interestingly, ALOS rises by **4%** for patients arriving between *00:00 am and 08:00 am*. For critical care pathway 2 a reduction in ALOS of **6.9%**, **14.1%**, and **6.8%** are projected across the three-time phases. Patients going through the critical care pathway 3 are projected to

**Table 4. ALOS ED patients face depending on care venue, patient pathway, arrival time for combined interventions policy where all the interventions—i.e., co-location, capacity of doctors, and observation ward and laboratory—are implemented simultaneously.**

| Policy 4 | Critical care pathways (minutes) | | | | | | | | | | | |
|---|---|---|---|---|---|---|---|---|---|---|---|---|
| | Pathway 1 | | | Pathway 2 | | | Pathway 3 | | | Pathway 4 | | |
| | Ph1 | Ph2 | Ph3 | Ph1 | Ph2 | Ph3 | Ph1 | Ph2 | Ph3 | Ph1 | Ph2 | Ph3 |
| BAU | 25 | 39 | 27 | 72 | 85 | 73 | 149 | 162 | 150 | 195 | 209 | 197 |
| 10% | 26 | 31 | 26 | 67 | 73 | 68 | 137 | 142 | 137 | 178 | 184 | 179 |
| 20% | 26 | 29 | 26 | 63 | 66 | 63 | 124 | 128 | 124 | 161 | 165 | 162 |
| 30% | 25 | 28 | 26 | 58 | 60 | 58 | 112 | 114 | 112 | 144 | 147 | 144 |
| | Ambulatory care pathways (minutes) | | | | | | Isolation care pathways (minutes) | | | | | |
| | Pathway 1 | | | Pathway 2 | | | Pathway 1 | | | Pathway 2 | | |
| | Ph1 | Ph2 | Ph3 | Ph1 | Ph2 | Ph3 | Ph1 | Ph2 | Ph3 | Ph1 | Ph2 | Ph3 |
| BAU | 72 | 166 | 85 | 196 | 290 | 208 | 42 | 42 | 42 | 101 | 101 | 102 |
| 10% | 67 | 93 | 72 | 179 | 204 | 183 | 42 | 42 | 42 | 95 | 95 | 96 |
| 20% | 63 | 72 | 64 | 162 | 171 | 163 | 42 | 42 | 42 | 90 | 89 | 90 |
| 30% | 58 | 63 | 58 | 145 | 149 | 145 | 42 | 42 | 42 | 84 | 83 | 84 |

experience **8%**, **12.3%** and **8.6%** reduction in ALOS across the time phases. Lastly, patients going through critical care pathway 4 are projected to experience **8.7%**, **11.9%** and **9.1%** reduction in ALOS across the time phases. Under the 30% assumptions, patients going through the critical care pathways 1 to 4 are projected to experience greater reductions in ALOS as indicated in Table 4. For critical care pathway 1 a reduction in ALOS of **28.2%** is projected from *08:00 am to 04:00 pm* and of **3.7%** for *04:00 pm to 00:00 am*. The ALOS of patients arriving between *00:00 am and 08:00 am* remains unchanged compared to the BAU. For critical care pathway 2 a reduction of **19.4%**, **29.4%**, and **20.5%** is projected across the three-time phases; while that for critical care pathway 3 are **24.8%**, **29.6%** and **25.3%** respectively across the time phases. Lastly, patients going through critical care pathway 4 are projected to experience **26.1%**, **29.6%** and **26.9%** reduction in ALOS across time phases under policy 4.

For patients going through the ambulatory care venue, ambulatory care pathway 1 patients, under the 10% assumptions, are projected to experience a reduction in ALOS by **6.9%** from *00:00 am to 08:00 am*, **43.9%** from *08:00 am to 04:00 pm*, and **15.2%** from *04:00 pm to 00:00 am*; while the reduction in ALOS for ambulatory care pathway 2 patients is **8.6%** from *00:00 am to 08:00 am*, **29.6%** from *08:00 am to 04:00 pm* and **12%** from *04:00 pm to 00:00 am*. As is the case with all the policies, under the 30% assumptions, ALOS is projected to reduce much more compared to the 10% assumptions as indicated in Table 4.

Finally, for patients going through the isolation care venue, ALOS for isolation care pathway 1 patients is projected to remain unchanged relative to the BAU across all time phases. For isolation care pathway 2, under the 10% assumptions, ALOS is projected to decline by **5.9%**, **5.9%** and **5.8%** respectively across time phases, whereas under the 30% assumptions, ALOS is projected to decline by **16.8%**, **17.8%** and **17.6%** respectively across time phases.

## 6. Discussion

### 6.1. General remarks

ED crowding is a multifaceted issue and so far, many solutions have failed because they ignored or underestimated the dynamic complexity of the problem [5]. After thoroughly reviewing the literature on ED crowding [47], recommends to 'research systems-wide solutions on the basis of existing evidence and operations theory, with the aim of mitigating the risk/problem of crowding.' For this reason, in the present study, we analyzed ED crowding

from a systems thinking perspective to explicitly account for the problem's dynamic complexity caused by a web of interrelated influencing factors [48]. More specifically, we used SD to map and simulate the interrelations among variables affecting and affected by ED crowding. The resulting simulation model helps to better understand the dynamic nature of this phenomenon and it can serve as an effective decision support system because of its capability to test policy proposals *in silico*. This has the pivotal advantage that ED mangers can experiment with policy proposals and study their consequences in a risk-free environment.

Given the current arrival pattern of patients and the configuration of the hospital-based ED in Singapore, a patient seeking emergency care, with the exception of a few periods of the day, is highly probable to receive care within the target time of 4 hours if triaged to the critical care or ambulatory care units or within 4.5 hours if triaged to the isolation care unit. As expected, the longer the care pathway of the patient (which includes consultation, laboratory, and observation), irrespective of the care venue, the larger the ALOS. Patients triaged to the critical care unit have the shortest ALOS compared to ambulatory and isolation care patients. Policies that focus on decanting P4 and P3 patients to a GP clinic co-located within the ED are more likely to reduce the ALOS of ambulatory patients, since all the P4 and P3 patients are triaged to the ambulatory care unit. Likewise, a policy that increases the efficiency of patient transfer from the observatory ward in the ED to the hospital ward, as well as decreasing the waiting time of laboratory investigation is more likely to reduce the ALOS of patients whose care pathway includes the observation ward and laboratory investigations.

The observed results can be explained by the interaction between the implemented policies (i.e., co-location policy, optimal allocation of doctor's policy, and observation ward and laboratory policy) and available ED capacity. For example, as patients are decanted to GP clinics on arrival at the ED, the share of patients triaged for emergency care decreases; therefore, waiting time for consultation will decrease resulting in a drop in ALOS. The decline in consultation waiting time is due to the fall in the number of patients waiting for consultation. Thus, available resources (ED doctors) are able to care for fewer patients demanding ED services. In addition, raising the number of ED doctors (via the optimal allocation policy) increases the resources (ED doctors) available to provide care, hence a reduction in ALOS as consultation waiting time declines. Lastly, an efficient transfer of patients from the observation ward to the acute hospital wards, as well as reduction of waiting time for laboratory and investigation is expected to decrease the ALOS of patients whose pathway includes the observation ward and laboratory and investigation. These policies were implemented based on the identification of ED bottlenecks—waiting time for consultation, laboratory and investigation waiting time and observation ward waiting time. These were identified as bottlenecks due to the significant time patients spend in these venues, thus increasing the ALOS of ED patients.

## 6.2. Main findings of this study

Ambulatory care patients with priority 3 and 4 make up the lion's share of all attending patients in our ED under study. On average, 55% of all patients are triaged to the ambulatory care area. Consequently, even a small reduction in the number of incoming patients into the ambulatory care area has a significant impact on waiting times and ALOS. This key finding has policy implications. Overall, the finding suggests that a comprehensive ED system that anticipates inappropriate self-referral of patients and makes provisions for such patients—by transferring such patients to a GP clinic in the ED—is more likely to triage only appropriate ED patients for emergency care, thus reducing the pressure on available ED resources. The lesson from this finding is that, there is a significant proportion of ED patients who inappropriately self-refer to the ED. Policymakers should anticipate that behavior and either provide GP

care co-located at the ED or incentive non-emergency patients—by reducing the out-of-pocket-cost—to seek GP care first before coming to the ED.

In Singapore, a pilot intervention called *GP first*, which incentivizes patients with less serious conditions to see their GP first before going to the ED was shown to reduce the number of non-emergency patients seeking care at the ED. In addition, EDs should be incentivized (by sharing savings from non-emergency patients triaged to co-located GPs) to triage patients accurately—to prevent up-coding where patients are assigned to higher severity than the actual condition to justify their use of ED services—and to ensure that patients are right-sited for care, that is, patients receive care at the appropriate venue with the lowest cost. The implication of this finding is that if EDs are inappropriately incentivized referring to the triage function, e.g., punished for providing ED care to non-emergency patients (P4), the EDs are likely to up-code non-emergency patients preventing the opportunity to improve care efficiency and reduce cost. Lastly, it is important to emphasis that, a sustainable approach to reduce ED crowding will require a well-functioning enhanced primary care system that improves health outcomes of the population and significantly lessens ED care demand among non-emergency patients. This is vital for countries with an aging population where demand for healthcare services is expected to increase. If the primary care system is not strengthened to provide appropriate care for the elderly population with multiple chronic diseases, inappropriate demand for ED care is expected to increase with its consequences of ED crowding.

Given that the main bottlenecks in most EDs are significant waiting times for consultation, laboratory investigation, and for hospital admission (mostly through the observation ward), it is important to ensure that proactive and innovative interventions are explored to reduce waiting times in these locations of ED care. Interventions that focus on the optimal allocation of ED doctors (typically increasing their numbers) should be explored to reduce consultation waiting time. In addition, efficient operating systems that ensure speedy transfer of patients from the observation ward in the ED to hospital wards should be implemented. For instance, interventions that focus on (i) categorization of wards to medical specialty; (ii) instituting a no reject policy; and (iii) performing ward level audits have been shown to improve waiting time for hospital admission [49].

## 6.3. Limitations of this study

The model presented here has some limitations. First, the use of an SD modeling approach for modeling ED patient flows introduces patient mixing that makes it difficult to track each patient individually; however, there are other modeling forms—such as agent-based modeling (ABM)—that focus on simulating the actions and interactions of autonomous agents that address this limitation. Patient mixing and the assumption that patients triaged into the same category have similar characteristics is an oversimplification that may affect the results. Second, transfer of patients to other hospitals—which was not relevant in our case—was not included in the model; whereas the likely impact of nurses and other allied health workers on waiting time was not included. Third, modeling results are reported as single values (ALOS) without an indication of the statistical uncertainty for the different venues of care and time phases. This could be improved by deriving confidence intervals for the modeling results through Monte Carlo simulation. Despite these limitations, the model presented herein remains useful for policymakers to test and evaluate innovative policies. For instance, the model could be used as an exploratory tool to search for high-leverage policies and to evaluate the likely impact of alternative policies on specific outcomes of interest. In addition, the model could help policymakers to design and communicate policy insights to stakeholders to build consensus and inform policy implementation.

## 7. Conclusions

This paper provides a detailed simulation model structure of patients' flows in a hospital-based ED in Singapore allowing for the exploration and evaluation of policies. The insights generated from the policy experiments suggest that to reduce ED crowding an enhanced primary care system is required. A strengthened primary care system has the potential to improve health outcomes of the population and, as a consequence, to reduce the demand for non-emergency care at the ED.

In view of this result, policymakers should design a cost-effective way to enhance primary care, co-locate GP clinics in all EDs to decant non-emergency patients seeking care at the ED, as well as incentivize all EDs to accurately triage patients and to send patients to the appropriate venues for care.

## Supporting information

**S1 Table. Summary table of input parameters.**
(DOCX)

**S2 Table. Variable list with full names and abbreviations.**
(DOCX)

**S1 Fig. Average length of stay (ALOS) for ED patients depending on care venue, patient pathway, arrival time for co-location policy where 10%-30% of P4 and P3 patients are decanted from the ED to a GP clinic co-located at the ED.**
(DOCX)

**S2 Fig. Average length of stay (ALOS) for ED patients depending on care venue, patient pathway, arrival time for optimal allocation of doctor's policy where the current ED doctors allocated are increased from 10%-30%.**
(DOCX)

**S3 Fig. Average length of stay (ALSO) for ED patients depending on care venue, patient pathway, arrival time for observation ward and laboratory waiting time policy where waiting times at the observation ward, and laboratory are reduced by 10%-30%.**
(DOCX)

**S4 Fig. Average length of stay (ALOS) for ED patients depending on care venue, patient pathway, arrival time for combined interventions policy where all the interventions—i.e. co-location, optimal allocation of doctors, and observation ward and laboratory—are implemented simultaneously.**
(DOCX)

## Author Contributions

**Conceptualization:** John Pastor Ansah, Salman Ahmad, Yuzeng Shen, Marcus Eng Hock Ong, David Bruce Matchar, Lukas Schoenenberger.

**Data curation:** John Pastor Ansah, Salman Ahmad, Lin Hui Lee, Yuzeng Shen, Marcus Eng Hock Ong, Lukas Schoenenberger.

**Formal analysis:** John Pastor Ansah, Salman Ahmad, Lin Hui Lee, Yuzeng Shen, Lukas Schoenenberger.

**Investigation:** John Pastor Ansah, Yuzeng Shen, Marcus Eng Hock Ong, Lukas Schoenenberger.

**Methodology:** John Pastor Ansah, Salman Ahmad, Lin Hui Lee, Yuzeng Shen, Marcus Eng Hock Ong, David Bruce Matchar, Lukas Schoenenberger.

**Project administration:** Yuzeng Shen, Marcus Eng Hock Ong, David Bruce Matchar, Lukas Schoenenberger.

**Resources:** Yuzeng Shen.

**Software:** John Pastor Ansah, Lin Hui Lee.

**Supervision:** John Pastor Ansah, Marcus Eng Hock Ong, Lukas Schoenenberger.

**Validation:** Marcus Eng Hock Ong, Lukas Schoenenberger.

**Visualization:** John Pastor Ansah, Salman Ahmad, Lukas Schoenenberger.

**Writing – original draft:** John Pastor Ansah, Salman Ahmad, Lin Hui Lee, David Bruce Matchar, Lukas Schoenenberger.

**Writing – review & editing:** John Pastor Ansah, Salman Ahmad, Yuzeng Shen, Marcus Eng Hock Ong, David Bruce Matchar, Lukas Schoenenberger.

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
