## [Decision Letter · Decision Letter 0]

11 Aug 2020

PONE-D-20-14680

Modeling Emergency Department Crowding: Restoring the Balance between Demand for and Supply of Emergency Medicine

PLOS ONE

Dear Dr. Schoenenberger,

Thank you for submitting your manuscript to PLOS ONE. After careful consideration, we feel that it has merit but does not fully meet PLOS ONE’s publication criteria as it currently stands. Therefore, we invite you to submit a revised version of the manuscript that addresses the points raised during the review process.

We look forward to receiving your revised manuscript.

Kind regards,

Yong-Hong Kuo

Academic Editor

PLOS ONE

Additional Editor Comments:

This manuscript has been reviewed by two experts in this area. Overall, they believe that the manuscript has potential to be published at PLOS ONE. However, there are still major concerns to be addressed before the manuscript can be considered for publication. Please refer to the Reviewers' comments to Author for the detailed comments.

Both reviewers also wish that the contributions of the research can be better presented. As the reviewers suggest, the literature on ED overcrowding is huge. Some recent significant related studies are suggested for review:

Ghanes, K., Wargon, M., Jouini, O., Jemai, Z., Diakogiannis, A., Hellmann, R., ... & Koole, G. (2015). Simulation-based optimization of staffing levels in an emergency department. Simulation, 91(10), 942-953.

Hu, X., Barnes, S., & Golden, B. (2018). Applying queueing theory to the study of emergency department operations: a survey and a discussion of comparable simulation studies. International transactions in operational research, 25(1), 7-49.

Kuo, Y. H., Chan, N. B., Leung, J. M., Meng, H., So, A. M. C., Tsoi, K. K., & Graham, C. A. (2020). An Integrated Approach of Machine Learning and Systems thinking for Waiting Time Prediction in an Emergency Department. International Journal of Medical Informatics, 104143.

Kuo, Y. H., Rado, O., Lupia, B., Leung, J. M., & Graham, C. A. (2016). Improving the efficiency of a hospital emergency department: a simulation study with indirectly imputed service-time distributions. Flexible Services and Manufacturing Journal, 28(1-2), 120-147.

Uriarte, A. G., Zúñiga, E. R., Moris, M. U., & Ng, A. H. (2017). How can decision makers be supported in the improvement of an emergency department? A simulation, optimization and data mining approach. Operations Research for Health Care, 15, 102-122.

Vanbrabant, L., Braekers, K., Ramaekers, K., & Van Nieuwenhuyse, I. (2019). Simulation of emergency department operations: A comprehensive review of KPIs and operational improvements. Computers & Industrial Engineering, 131, 356-381.

Vanbrabant, L., Martin, N., Ramaekers, K., & Braekers, K. (2019). Quality of input data in emergency department simulations: Framework and assessment techniques. Simulation Modelling Practice and Theory, 91, 83-101.

Yousefi, M., Yousefi, M., & Fogliatto, F. S. (2020). Simulation-based optimization methods applied in hospital emergency departments: A systematic review. Simulation, 0037549720944483.

Journal Requirements:

2. We note you have included a table to which you do not refer in the text of your manuscript. Please ensure that you refer to Table 2 and 3 in your text; if accepted, production will need this reference to link the reader to the Table.

Reviewers' comments:

Reviewer's Responses to Questions

**Comments to the Author**

1. Is the manuscript technically sound, and do the data support the conclusions?

Reviewer #1: Yes

Reviewer #2: Partly

2. Has the statistical analysis been performed appropriately and rigorously? 

Reviewer #1: Yes

Reviewer #2: I Don't Know

3. Have the authors made all data underlying the findings in their manuscript fully available?

Reviewer #1: Yes

Reviewer #2: No

4. Is the manuscript presented in an intelligible fashion and written in standard English?

Reviewer #1: Yes

Reviewer #2: Yes

5. Review Comments to the Author

Reviewer #1: This paper presented a comprehensive simulation model built for a hospital-based emergency department (ED) in Singapore using system dynamics (SD). The model was then used to test the performance of three policies intended to streamline patient flow in the ED and reduce crowding. Policy 1 works by decanting lowest-priority patients from the ED to the general practitioner (GP) clinic. Policy 2 adds to Policy 1 by enforcing a 20% reduction on the waiting time in the observation ward of patients to be transferred to the main hospital. Policy 3 improves upon both Policy 1 and 2 by adding 20% staffing capacity. The main finding shows that Policy 1 most significantly decrease patient average length-of-stay (ALOS).

The model has thoroughly captured all major patient flow mechanisms in the three primary divisions within the ED system, i.e., critical care area, isolation area, and ambulant area, and meticulously discussed in simple mathematical terms. However, I have several major concerns about the methodology as well as some of the assumptions adopted by the authors.

First, the model assumed constant sojourn time for the majority of the service processes such as waiting time (for the providers), consultation time (with the providers), time spent waiting for lab results, and time spent in the observation ward. The authors frankly acknowledged these assumptions as limitations yet provide very little justification or ramification for them. In my opinion, it is critical to capture the time- and census- dependency of these sojourn times to accurately reflect the potential improvement of a policy on the real-world ED system. It was mentioned that the authors had access to patient arrival data and conducted an additional observational study to obtain the process timings. Thus, I would suggest them to utilize the arrival and departure time stamps in the data to impute empirical distributions for the corresponding sojourn times. This effort, if feasible, should increase the robustness and credibility of the results.

Second, there seems to be room for improvement for the policies examined. To be more specific, Policy 2 imposed a 20% reduction on boarding time (time spent in the observation ward waiting to be transferred to the main hospital). I wonder where this value comes from – whether it is realistic, and if so, how it can be achieved. Also, instead of presenting an isolated value I would recommend testing a few more scenarios, e.g., from 5% to 50% with 5-10% increment. Although I hold doubt about the feasibility of reaching a time invariant reduction in boarding time, it would be great to see some explanations for the choice of this value, as well as some concrete examples of how it can be achieved in a real ED system.

My concern with Policy 3 can be summarized in a similar vein. I.e., there seems to be a lack of rationale behind the single choice of 20% for the increase in physician capacity. Moreover, the authors addressed this policy as dynamic. I could be missing something here, but it seemed to be a static policy in that the staffing capacity was not dependent on the system state (time or census level). It would be interesting to see the authors incorporating a few more sophisticated policies targeting at unclogging the bottleneck of the queues, as well as providing some discussions around how the target reduction of demand or increase of supply can be obtained in reality.

To my knowledge, the majority of the literature on ED simulation utilizes discrete-event simulation, the advantage of which over SD is that one can track the system state and obtain statistics at individual patient level. As a result, it is a drawback that one cannot evaluate the policies’ impact on ALOS by patient triage priority, especially given the fact that the bottleneck of ED crowding usually lies in the non-critical patients consisting the majority of the arrivals. Still, I think this paper does bring in novelty by exploring ED dynamics through the lens of SD. I personally have very limited knowledge of this simulation technique and have not used it for my own research. Yet, upon a short literature review, I see that SD does offer some appealing features: First, the model structure can be explained and presented in simple mathematical terms, which is suitable to be communicated to non-technical audience; Second, the model takes high-level policies as inputs, which again, makes it accessible for interpretation and necessitates dialogue between hospital stake holders and the modeling team. Furthermore, the major finding of this paper coincides with the landscape of policy reforms addressing ED crowding in Singapore. As the authors pointed out, the fact that a simple policy of decanting lowest priority patients from ED to GP clinics significantly reduces ALOS serves as a strong backbone for the country’s “GP first” movement. I appreciate that the authors are taking the pioneering step in tackling the problem of simulating EDs using a SD perspective and I think that after some refinement this work can serve as a motivation and basis for future work on diversifying ED simulation techniques and alternative policy development and implementation.

Finally, I would like to outline the rest of my minor comments/recommendations as below:

1. Add a section for input parameter analyses and present estimated values for the key processing times such as triage time, registration time, waiting time to see a physician, consultation with a physician, waiting time to be transferred, etc.

2. Provide definition of AT (adjustment time) in the paper and short explanations for its use in the equations.

3. In my humble opinion, it would be sufficient and possibly serve better visual purpose to present smoothed hourly average ALOS instead of the histogram of raw simulation data in Figure 10 and 11.

4. In Figure 10 and 11, it would be good to use more distinctively different colors for base case scenario and target time horizontal line.

5. Since in some scenarios policy 1 and 2 had no effect on reducing ALOS it would be less confusing to use the same legend or add a footnote to the bottom of Figure 10 to highlight this part of the results.

I hope the authors find my comments helpful and wish the authors the best in progressing this research.

Reviewer #2: In this paper, a system dynamics simulation model of an ED is developed and extensively described. The simulation model is used to investigate three policies to reduce the impact of ED crowding on ED performance. Although the topic of the paper is interesting, the paper needs a major revision before it can be considered for publication in PLOS ONE. Special attention should be given to the positioning/communication of the scientific contribution of the paper. It is not clear whether the main contribution and novelty of this study lies in the development of a system dynamics simulation model, in the investigation of the three policies to improve ED performance, or both. In addition, the main part of the paper deals with the mathematical description of the system dynamics simulation model, while the sections dealing with the experiments and results are rather short. A more thorough explanation of the base case setting in the ED under study, the experiments and the results can provide added value to the paper. While I believe the authors have done a lot of relevant work, they fail at this moment to convince the reader of the main academic and practical contributions of their research.

The following remarks and comments can be made for each of the paper sections:

- Reading the Abstract, the actual contribution and main conclusion of the paper is not clear. The authors should explicitly state the novelty and main insights of their work

- The Introduction section fails to clarify the academic and practical contribution of the paper. What has already has been done in the field? How does this paper differ from existing research? What is its specific added value, both for theory and practice?

o p2, last paragraph: “Typically, patient arrival patterns are cyclic … This variable demand poses an additional burden on ED management teams.”. The fact that arrival patterns are cyclic doesn’t necessarily result in a more difficult management of EDs, as the same pattern can be repeated every time. However, in addition to the cyclic pattern, patient arrivals are unpredictable and stochastic, which makes ED management more difficult.

o p2, last paragraph: “Although the number of emergency physicians (EPs) has risen, that is, 13.4% annually…”. Can it be explained why there is a high physician workload and ED crowding, while the number of ED physicians has risen more than double as fast as the number of ED patient visits? One of the investigated policies focuses on the capacity of ED physicians, but are they really the bottleneck resource?

o Figure 2: The word “quite” should be replaced by “quiet” in the caption.

o p4, first paragraph: “This allows identifying bottlenecks in the system, i.e., ED venues where patients accumulate and put a strain on ED performance.”. The authors define bottlenecks as areas in the ED, rather than specific ED resources. However, resolving a bottleneck which is a complete area in the ED is difficult to accomplish and requires further examination of the specific problem areas within the venue. In addition, this paragraph suggests that bottlenecks will be identified by means of the simulation model, which is not the case in the paper.

o p4, first paragraph: The description of the three investigated policies is a little bit misleading. For example, it is stated that the paper will investigate “How ED manpower needs to be adjusted in order to dissolve congestions and smooth ED patient flows”, while only a 20% increase in physician capacity for all venues at all times is investigated, without a prior identification of the bottleneck resources/venues/times and accompanying required adjustments in ED manpower.

o p4, last paragraph: “To the best of our knowledge, there is no study using SD on the development of a virtual ED, as we understand the term―a simulation model that comprehensively reflects all major patient flows and medical resources in an ED―that is fully transparent (documented) and accessible for researchers and subject experts.”. Is this the main contribution of the paper? And to what extent is the developed simulation model generic and reusable by other researchers, as the structure of the simulation model is based on the ED under study (e.g. care pathways, venues…)?

- The value of the Literature Review section is limited, for the following reasons:

o Only six studies in the vast amount of ED simulation literature are described, while the authors indicate that a recent literature review paper (Salmon et al., 2018) identified 18 studies that applied system dynamics to EDs. Why only focus on these six studies? What is the link/difference between the described studies and this paper? What are the shortcomings of these studies and how are they dealt with in the current study?

o Some references are wrongly placed between brackets in this sections, e.g. (Lane et al., 2000).

o p5, first paragraph of Literature Review section: “… (the rationale for choosing SD is explained in the study setting section below)”. As this is the second time the authors refer to the study setting section for the rationale behind choosing SD, a short explanation can already be provided here to justify this choice and to clarify the contributions compared to existing literature. Why is it important to use SD in addition to all DES studies? What are the advantages?

o It is not clear how this paper contributes to the existing (and large) body of literature. What is the contribution to existing literature and what is link between existing literature and this research? In case the main contribution of this paper lies within the use of system dynamics to model an ED, a more thorough discussion of the advantages of SD, existing SD models of EDs, and the novelties within this model is required (see previous comment). In case the main contribution are the investigated policies and accompanying insights, an overview of existing studies that examine the three policies investigated in this paper can provide useful information on the added value of this paper compared to existing studies.

- From the Methods section (Section 3.1), it does not become clear what the advantages of SD are. Why is it important to also use other methods than DES, and especially SD? Which new insights can SD provide? The use of a diversity of simulation methods is only valuable when all the different methods have their advantages (which is the case, but does not become clear from the text).

- In Section 3.2 on the overall structure of the ED, some more information regarding the ED under study and the characteristics of the ED would be interesting, as there exists no general structure that represents every ED, and the effectiveness of improvement policies depends on ED characteristics (e.g. yearly number of patient arrivals, number of physicians working in each venue/shift, number of beds in the ED, percentages of patients per venue…). This might also enhance the understanding of the experiments and results presented in sections 4, 5 and 6.

o p8, first paragraph: “The higher the ED patient’s acuity or priority, the greater the average physician consultation time.”. How is this determined, based on real-life data, observations, literature?

o Figure 3: This figure also contains ‘treatment’ and ‘follow-up consultation’ as part of the patient flow through the ED. Are these also included in the simulation model? Based on the care pathways discussed in Section 3.3, all patients seem to have only one consultation with a physician, which does not represent the actual patient flow in most EDs.

- The Model Structure section (Section 3.3) is difficult to read and understand because of the many equations and abbreviations. Depending on the main contribution of the paper (i.e. system dynamics or policy investigation), this section can be shortened and some equations can be placed in the appendix. For example, the four care pathways in the critical care area (Section 3.3.2) are all possible pathways in the ED (i.e. no new pathways are introduced in the other two zones). As the pathways and accompanying mathematical equations are discussed in great detail in Section 3.3.2, the description in Sections 3.3.3 and 3.3.5 can be shortened by indicating that the reasoning and equations are comparable to the pathways in the critical care area, but with other parameter values. In addition, the abbreviations make it difficult to understand the mathematical equations and model description. Would it be possible to provide an explanation of all the stocks and flows in words, in addition to the mathematical equations? Especially for non-experts (e.g. ED staff) this might be useful to better understand the model.

o p9, first paragraph: “… developed from a broad range of empirical data.”. What empirical data is used in this study? As the appendix contains a table with all model parameters and their source, a reference to this table should be added to the text.

o p13, second paragraph: “… AT is adjustment time …”. What is meant with adjustment time?

o p14, equation (33): As POW refers to all patients under observation in the waiting area, shouldn't this equation only contain the fraction of patients that is discharged home, because POW also contains admitted patients and these are not discharged home? An explanation in words would enhance the understandability of the equations.

o p14, Section 3.3.3: Why is ‘AB’ added to all abbreviations to refer to ambulatory care (and IS to refer to isolation care in Section 3.3.5), while no reference to critical care is added to the abbreviations in Section 3.3.2? Try to be consistent, as this will enhance the understanding of the many abbreviations.

- The Data sources section (Section 3.4) is very short, especially as very little information is provided within the other sections on the characteristics of the ED under study, the data used to determine model parameters and the values of these parameters.

o p22, last paragraph: “The patient arrival data used in the simulation model was the highest patient arrival pattern during the period of June – August 2017 …”. Does this arrival pattern consist of just one day of patient arrivals? Or one month? What are the run parameters of the simulation model? And why only using the highest arrival pattern, is this representative for actual ED operations?

o P23, last sentence of Section 3.4: “The simulation model is provided in the supplementary materials for review.”. There was no supplementary material available regarding the simulation model.

- In the Model validation section (Section 3.5), a more comprehensive explanation of the statistical tests is required to better understand the validation process. How to interpret MAPE, and how to determine whether the MAPE value indicates a good fit between the simulation model and actual data? The same for the Theil statistic, how to interpret the U-values and how to determine whether they represent a good fit?

o p23, second paragraph of section 3.5: The behavior validation tests only look at the average length of stay across all venues. However, when the variability in LOS within and across venues (and time periods) is high, the average LOS of the simulation model and data can be comparable, while the simulation model poorly represents actual ED operations.

o p23, last sentence: “This suggests that the simulated variables track the underlying trend well, but diverge when comparing point-by-point, indicating that majority of the errors are unsystematic with respect to the purpose of the model.”. Explain? And what does the difference in U-values between the critical care area, ambulatory care area and isolation care area mean? Does the model only provide a good fit for some of the areas?

- Regarding the Policy Experiments section, some information on the reason for choosing these policies is lacking. What is the current ED performance and what are the bottlenecks? How are the experiments determined in order to resolve these bottlenecks?

o Base case scenario: In the text, this scenario is referred to as a ‘hypothetical scenario’, but isn’t this the current ED setting?

o Policy 3: An increase by 20 percent seems a very high increase, given the fact that the number of patients in the ED decreases through co-location. Is an equal increase in the number of ED physicians required in all venues and at all times (e.g. the average LOS is higher in phase 2 than in the other phases)? In the results section, it appears that the introduction of additional physician capacity is not capable of reducing LOS for all patients in the ED.

- The Results section contains a lot of interesting results, but an explanation of the results is lacking. What explains the large differences in LOS between different care pathways and time intervals? Why are certain policies more effective to reduce LOS? And why do some care pathways benefit more from a certain policy? In addition, are the improvements in LOS statistically significant?

o Based on the results in Table 1, policy 1 seems least effective. Why do the authors only focus on this policy in the conclusion section?

o Figure 10: The base case scenario seems lacking on these graphs.

o Figure 10 and 11: How can the large fluctuations in LOS be explained (even within a time period/phase)?

o Figure 11, isolation pathways: The target time is not presented on the graph regarding isolation pathway 1, probably because of the scale on the y-axis. Why is the scale on the y-axis different on both graphs of the isolation pathways?

- In the Discussion and Conclusions sections, the focus is on the insights regarding the co-location of general practitioners in the ED. I agree that this is a very interesting improvement option to reduce ED crowding, but are there no main insights regarding the other policies and the SD model? The conclusions should provide an overview of the main contributions of the paper, and based on the current conclusions the only contribution seems to be the investigation of the co-location of general practitioners in the ED. If this is the main contribution, the introduction and literature review section should be rewritten to focus on this contribution (e.g. what literature does already exist on the use of GPs in an ED, and what is the added value of this study compared to existing literature?). Furthermore, a critical reflection on how the investigated policies can be implemented in practice, and on the generalizability of the results, is lacking.

o p29, last paragraph: “In addition, raising the number of ED doctors (via the dynamic allocation policy) …”. The investigated policy seems no dynamic allocation policy, as only a 20% increase in physician capacity for all venues and time periods is investigated.

o p30, first paragraph of Section 6.2: “The key finding that decanting non-emergency patients who seek care at the ED to a GP clinic co-located at the ED significantly decreases ALOS and improves patient experience has policy implications.” Is this the key finding? Based on the results in Table 1, this policy seems least effective to reduce LOS.

o p30, first paragraph of Section 6.2: “The lesson from this finding is that, there is a significant proportion of ED patients who inappropriately self-refer to the ED.” Is this true? The policy investigates the effect of referring 50% of the P3 and P4 patients to a GP, but does 50% of these patients inappropriately go to the ED for care? How is this percentage determined?

o Section 6.3: The advantages of using SD do not become clear from this section (e.g. SD results in an oversimplification of actual ED operations).

o Section 7: A paragraph on future research opportunities is lacking.

6. PLOS authors have the option to publish the peer review history of their article (what does this mean?). If published, this will include your full peer review and any attached files.

Reviewer #1: No

Reviewer #2: No

---

## [Author Response · Author response to Decision Letter 0]

31 Oct 2020

Dear editor, dear reviewers,

we have uploaded our responses to your requests in a separate file. Thanks for your time and effort. Best

---

## [Decision Letter · Decision Letter 1]

19 Nov 2020

PONE-D-20-14680R1

Modeling Emergency Department Crowding: Restoring the Balance between Demand for and Supply of Emergency Medicine

PLOS ONE

Dear Dr. Schoenenberger,

Thank you for submitting your manuscript to PLOS ONE. After careful consideration, we feel that it has merit but does not fully meet PLOS ONE’s publication criteria as it currently stands. Therefore, we invite you to submit a revised version of the manuscript that addresses the points raised during the review process.

We look forward to receiving your revised manuscript.

Kind regards,

Yong-Hong Kuo

Academic Editor

PLOS ONE

Additional Editor Comments (if provided):

The revision has been reviewed by one of the reviewers from the last round. The other reviewer was unable to accept the review invitation. I have gone through the revision and reviewer's comments. I suggest minor revision.

Reviewers' comments:

Reviewer's Responses to Questions

**Comments to the Author**

1. If the authors have adequately addressed your comments raised in a previous round of review and you feel that this manuscript is now acceptable for publication, you may indicate that here to bypass the “Comments to the Author” section, enter your conflict of interest statement in the “Confidential to Editor” section, and submit your "Accept" recommendation.

Reviewer #2: (No Response)

2. Is the manuscript technically sound, and do the data support the conclusions?

Reviewer #2: Yes

3. Has the statistical analysis been performed appropriately and rigorously? 

Reviewer #2: I Don't Know

4. Have the authors made all data underlying the findings in their manuscript fully available?

Reviewer #2: Yes

5. Is the manuscript presented in an intelligible fashion and written in standard English?

Reviewer #2: Yes

6. Review Comments to the Author

Reviewer #2: The authors are to be commended for revising the manuscript. The academic and practical contribution of the paper have been clarified, and the experiments and results sections are extended and rewritten, which enhances the quality of the paper.

Following are some comments that still need to be addressed:

Section 2

- The contributions of the paper are clarified in the introduction, but the added value of the literature review section is still rather limited. As indicated by the authors, an extensive literature review is not the goal of the paper and also not required. However, it is indicated that 18 papers exist on the application of SD in an ED context, while only 6 papers are discussed. At least, an explanation should be provided for only discussing these 6 papers in detail: Why are these papers most relevant to be discussed? In addition, a short discussion on how the current paper relates to the existing (and discussed) literature should be included at the end of the section to position the paper within the state-of-the-art literature.

Section 3

- The term ‘adjustment time’ should be clarified within the paper. In the response to review comments (comment 19) an explanation is provided, but this explanation should also be included in the paper such that readers understand this term.

- Section 3.5, p23: “The patient arrival data used in the simulation model was the highest patient arrival pattern during the period…”. In the response to reviewer comments (comment 23), it became clear that a 24 hour arrival pattern corresponding to a peak day in patient arrivals was used as input to the simulation model, in order to clearly represent the operational problems in the ED. However, this is still not clear from the text, so it might be interesting to include the explanation of the response to reviewer comments in the text.

- Section 3.5: Patient records and observations are indicated as the two data sources, but expert opinion also seems an important source of input data based on S1 Table 5 and is currently not mentioned in this section.

- Section 3.7: This section clearly adds value to the paper, but it can be integrated with Section 3.5, since both sections are rather short and deal with the use of input data to determine simulation model parameters.

- Section 3.6: The explanation of the statistics makes this section more clear, but in my opinion an explanation for the difference in Theil statistic value between the different care areas is still lacking. Why is there a difference between the care areas? And since the majority of errors for the ambulatory care area does not come from the covariation component, is the simulation model a good representation of actual operations in the ambulatory care area?

Section 4

- p26, policy 3: “This policy experiment gradually increases the number of ED doctors currently allocated to CCA, ambulatory care and isolation care from 10% to 30%...” This sentence is misleading in two different ways: (1) Based on this sentence, it seems that the number of allocated doctors is currently at 10% and gradually increases to 30%, but the base case scenario is a 0% increase; (2) In the next paragraphs, it becomes clear that the capacity of doctors is gradually increased by 10%, 20% and 30%, but based on this sentence it seems that the number of allocated doctors varies between 10% and 30%, but then the reader might pose the question: 10% or 30% of which doctors is allocated to the ED? Of the total amount of doctors in the hospital?. In order to avoid confusion, ‘Optimal capacity of doctors’ might be a better name for the policy, since the capacity of doctors is increased (no change in allocation of doctors between hospital departments).

Section 5

- Sometimes not all time phases are discussed in the description of the results

o Section 5.2: The results for the ambulatory care area are only discussed for time phases 2 and 3, not for time

phase 1 (00:00am- 08:00am).

o Section 5.5: Only 08:00am – 04:00pm is discussed for critical care pathway 1 under the 30% scenario,

00:00am-08:00am is lacking for ambulatory care pathway 1 under the 10% scenario.

- A lot of results are presented in this section, but an explanation of the results is sometimes lacking. An explanation can enhance the understanding of the results and support the main insights from the experiments.

o Section 5.2: Why is there no impact of policy 1 on isolation care pathways, since there are also P3 and P4

patients in this care area?

o Section 5.3: Why is there no influence on the ALOS of isolation care patients? The capacity of physicians also

increased in this area?

o Section 5.4: There seems to be only an influence on the ALOS for care pathways that include observation, not

for care pathways that include laboratory examinations without observation. How can this be explained, as the

waiting time for laboratory examinations is also reduced?

- Are the results regarding the comparison of the different policies with the BAU scenario statistically significant? In simulation there is a lot of randomness, which can for example be seen in the graphs of S1 Fig 10-13, so is it possible to compare the results of different policies only based on a mean value without statistical tests?

Section 6

- Section 6.2: From the text, it is still not clear why the insights regarding the co-location policy are the key findings of the paper. Other policies also seem efficient for several types of patients, sometimes even more efficient as they result in a higher reduction in ALOS. In the response to reviewer comments (comment 38), it is indicated that ambulatory care patients with priority P3 and P4 are the large majority of patients in the ED, and a small reduction in the number of patients from this category already highly impacts ED performance. This explanation should be included in this section to explain why this is indicated as the key finding of the study.

7. PLOS authors have the option to publish the peer review history of their article (what does this mean?). If published, this will include your full peer review and any attached files.

Reviewer #2: No

---

## [Author Response · Author response to Decision Letter 1]

1 Dec 2020

Dear reviewer, please find our reply to your comments at the end of our submitted document. Thanks for your time and effort. Warm regards

---

## [Decision Letter · Decision Letter 2]

3 Dec 2020

Modeling Emergency Department Crowding: Restoring the Balance between Demand for and Supply of Emergency Medicine

PONE-D-20-14680R2

Dear Dr. Schoenenberger,

We’re pleased to inform you that your manuscript has been judged scientifically suitable for publication and will be formally accepted for publication once it meets all outstanding technical requirements.

Kind regards,

Yong-Hong Kuo

Academic Editor

PLOS ONE

Additional Editor Comments (optional):

Based on the reviewer's recommendation and comments, the comments from the last round have been successfully addressed and so I recommend accept.

Reviewers' comments:

Reviewer's Responses to Questions

**Comments to the Author**

1. If the authors have adequately addressed your comments raised in a previous round of review and you feel that this manuscript is now acceptable for publication, you may indicate that here to bypass the “Comments to the Author” section, enter your conflict of interest statement in the “Confidential to Editor” section, and submit your "Accept" recommendation.

Reviewer #2: All comments have been addressed

2. Is the manuscript technically sound, and do the data support the conclusions?

Reviewer #2: Yes

3. Has the statistical analysis been performed appropriately and rigorously? 

Reviewer #2: Yes

4. Have the authors made all data underlying the findings in their manuscript fully available?

Reviewer #2: Yes

5. Is the manuscript presented in an intelligible fashion and written in standard English?

Reviewer #2: Yes

6. Review Comments to the Author

Reviewer #2: (No Response)

7. PLOS authors have the option to publish the peer review history of their article (what does this mean?). If published, this will include your full peer review and any attached files.

Reviewer #2: No

---

## [Editor Report · Acceptance letter]

7 Dec 2020

PONE-D-20-14680R2 

Modeling Emergency Department Crowding: Restoring the Balance between Demand for and Supply of Emergency Medicine 

Dear Dr. Schoenenberger:

I'm pleased to inform you that your manuscript has been deemed suitable for publication in PLOS ONE. Congratulations! Your manuscript is now with our production department. 

Kind regards, 

on behalf of

Dr. Yong-Hong Kuo 

Academic Editor

PLOS ONE